# A choreography of centrosomal mRNAs reveals a conserved localization mechanism involving active polysome transport

Adham Safieddine [1,2,3✉], Emeline Coleno[1,2], Soha Salloum[1,2,3,11], Arthur Imbert[4,5,6,11], Abdel-Meneem Traboulsi[1,2], Oh Sung Kwon[7], Frederic Lionneton[8], Virginie Georget [8], Marie-Cécile Robert[1,2], Thierry Gostan[1], Charles-Henri Lecellier [1,2], Racha Chouaib[1,2,3], Xavier Pichon[1,2], Hervé Le Hir[7], Kazem Zibara [3], Florian Mueller [9], Thomas Walter [4,5,6], Marion Peter [1,2] & Edouard Bertrand [1,2,10✉]

Local translation allows for a spatial control of gene expression. Here, we use high-throughput smFISH to screen centrosomal protein-coding genes, and we describe 8 human mRNAs accumulating at centrosomes. These mRNAs localize at different stages during cell cycle with a remarkable choreography, indicating a finely regulated translational program at centrosomes. Interestingly, drug treatments and reporter analyses reveal a common translation-dependent localization mechanism requiring the nascent protein. Using *ASPM* and *NUMA1* as models, single mRNA and polysome imaging reveals active movements of endogenous polysomes towards the centrosome at the onset of mitosis, when these mRNAs start localizing. ASPM polysomes associate with microtubules and localize by either motor-driven transport or microtubule pulling. Remarkably, the *Drosophila* orthologs of the human centrosomal mRNAs also localize to centrosomes and also require translation. These data identify a conserved family of centrosomal mRNAs that localize by active polysome transport mediated by nascent proteins.

---

[1] Institut de Génétique Moléculaire de Montpellier, University of Montpellier, CNRS, Montpellier, France. [2] Equipe Labélisée Ligue Nationale Contre le Cancer, University of Montpellier, CNRS, Montpellier, France. [3] ER045, PRASE, and Biology Department, Faculty of Sciences-I, Lebanese University, Beirut, Lebanon. [4] MINES ParisTech, PSL-Research University, CBIO-Centre for Computational Biology, Fontainebleau, France. [5] Institut Curie, Paris, Cedex, France. [6] INSERM, U900, Paris, Cedex, France. [7] Institut de Biologie de l'Ecole Normale Supérieure (IBENS), Ecole Normale Supérieure, CNRS, INSERM, PSL Research University, Paris, France. [8] BioCampus Montpellier, Montpellier, Cedex, France. [9] Imaging and Modeling Unit, Institut Pasteur, UMR 3691 CNRS, C3BI USR 3756 IP CNRS, Paris, France. [10] Institut de Génétique Humaine, University of Montpellier, CNRS, Montpellier, France. [11] These authors contributed equally: Soha Salloum, Arthur Imbert. ✉email: Adham.Safieddine@igmm.cnrs.fr; Edouard.Bertrand@igmm.cnrs.fr

Messenger RNA localization is a post-transcriptional process by which cells target certain mRNAs to specific subcellular compartments. The trafficking of mRNA molecules is linked to its metabolism and function[1]. Indeed, the subcellular localization of a transcript can influence its maturation, translation, and degradation. On one hand, mRNAs can be stored in a translationally repressed state in dedicated structures such as P-bodies[2,3]. On the other hand, some mRNAs can localize to be translated locally. Such a local protein synthesis can be used to localize the mature polypeptide, and in this case, it can contribute to a wide range of functions, such as cell migration, cell polarity, synaptic plasticity, asymmetric cell divisions, embryonic patterning, and others[4–6]. Recently, local translation has also been linked to the metabolism of the nascent protein, rather than to localize the mature polypeptide. This is for instance the case for mRNAs translated in distinct foci termed translation factories, which correspond to small cytoplasmic aggregates containing multiple mRNA molecules of a given gene[7,8].

Specific subcellular localization of mRNA molecules can be achieved by several mechanisms. Passive diffusion coupled with local entrapment and/or selective local protection from degradation are two strategies that can establish specific distributions of mRNA molecules[6]. In most cases, however, mRNA transport and localization occurs via motor-driven transport on the cytoskeleton[9–11]. Molecular elements that regulate and control mRNA localization include cis- and trans-acting elements. Cis-acting elements are referred to as zip codes and are often found within the 3′UTR of the transcript[12–14]. Many types of zip codes have been described based on primary sequence, number, redundancy, and secondary/tertiary structure. Zip codes are defined by their ability to carry sufficient information for localizing the transcript. They bind one or several trans-acting RNA-binding proteins (RBPs), which mediate diverse aspects of RNA metabolism such as motor binding and translational regulation[6]. Indeed, mRNAs in transit are often subjected to spatial control of translation[15]. A long-standing notion in the field is that the transport of localized mRNAs occurs in a translationally repressed state, which serves to spatially restrict protein synthesis[16,17]. Local translational derepression occurs once the transcript has reached its destination, for instance by phosphorylation events and/or competition with pre-existing local proteins[18–20].

While active transport of transcripts through RNA zip codes appears to be a frequent mechanism, mRNA localization can also involve the nascent polypeptide, as in the case of secreted proteins. Here, the signal recognition particle (SRP) binds the nascent signal peptide, inhibits translation elongation, and mediates anchoring of the nascent polysome to the SRP receptor on the endoplasmic reticulum, where translation elongation resumes[21–23]. Recently, a few hints, such as puromycin sensitivity, suggested that translation may play a role in the localization of some other types of mRNAs[8,24]. Whether this is indeed the case and the mechanisms involved remain, however, unknown.

Centrosomes are ancient and evolutionary conserved organelles that function as microtubule (MT) organizing centers in most animal cells. They play key roles in cell division, signaling, polarity, and motility[25,26]. A centrosome is composed of two centrioles and their surrounding pericentriolar material (PCM). In cycling cells, centriole duplication is tightly coupled to the cell cycle to ensure a constant number of centrioles in each cell after mitosis[27]. Briefly, G1 cells contain one centrosome with two centrioles connected by a linker. At the beginning of S phase, each parental centriole orthogonally assembles one new procentriole. This configuration is termed engagement and prevents reduplication of the parental centrioles. Procentrioles elongate as the cell is progressing through S and G2. In G2, the two centriolar pairs mature and PCM expands, in preparation of mitotic spindle formation[25]. The G2/M transition marks the disruption of the centriole linker and centrosome separation. The first clues suggesting the importance of mRNA localization and local translation at the centrosomes were discovered almost 20 years ago in *Xenopus* early embryos[28]. It was found that cyclin B mRNAs concentrated on the mitotic spindle and that this localization was dependent on the ability of cytoplasmic polyadenylation element-binding protein (CPEB) to associate with MTs and centrosomes. A more global view was obtained by purifying mitotic MT-bound mRNAs in *Xenopus* eggs and synchronized HeLa cells[29]. It was also shown that 3′UTR CPEs regulate the localized translation activation of spindle enriched mRNAs that is essential for the first meiotic division in *Xenopus* oocytes[30]. Another approach used microscopy and systematic in situ hybridization to image RNA localization in *Drosophila* embryos[31]. Although this study did not reach single-molecule sensitivity, it revealed that six mRNAs localized at centrosomes across different stages of early *Drosophila* development. In the following study, 13 mRNAs were annotated as enriched on *Drosophila* centrosomes in at least one stage or tissue over the full course of embryogenesis[32]. Finally, two natural antisense mRNAs (cen and ik2) were recently shown to co-localize to centrosomes in a co-dependent manner via 3′ UTR interactions in *Drosophila* embryos[33]. In humans, four mRNAs (*PCNT*, *HMMR*, *ASPM, and NUMA1*), and two translation factors (eIF4E and the phosphorylated ribosomal protein p-RPS6) were recently found to localize at centrosomes[8,24]. These mRNAs all code for centrosomal proteins, suggesting that they are translated locally.

Here, we performed a systematic single-molecule fluorescent in situ hybridization (smFISH) screen of almost all human mRNAs coding for centrosomal proteins and we described a total of eight transcripts localizing at centrosomes. Remarkably, all eight mRNAs required synthesis of the nascent protein to localize and, by imaging single *ASPM* and *NUMA1* mRNAs and polysomes, we demonstrate that localization occurs by active transport of polysomes. Moreover, the *Drosophila* orthologs of the human centrosomal mRNAs also localized to centrosomes in a similar translation-dependent manner. This work thus identifies a conserved family of centrosomal mRNAs that become localized by active polysome transport.

## Results

**Screening genes encoding centrosomal proteins reveals a total of eight human mRNAs localizing at the centrosome**. In order to acquire a global view of centrosomal mRNA localization in human cells, we developed a high-throughput smFISH technique (HT-smFISH) and screened genes encoding centrosomal and mitotic spindle proteins. The experimental pipeline is described in Fig. 1a. Briefly, we designed 50–100 individual probes against each mRNA of the screen. The probes were then generated from complex pools of oligonucleotides (92,000), first by using gene-specific barcode primers to polymerase chain reaction (PCR) out the probes of single genes, followed by a second round of PCR to add a T7 promoter and in vitro transcription to generate single-stranded RNA probes (Fig. 1a; see "Materials and methods"). The probes were designed such that each contained a gene-specific sequence flanked by two overhangs common to all probes (Flaps X and Y). A pre-hybridization step then labeled the overhangs with fluorescently labeled locked nucleic acid (LNA) oligonucleotides, and the heteroduplexes were hybridized on cells as in the smiFISH technique[34], except that cells were grown and hybridized on 96-well plates. This approach is cost-effective because the probes are generated from an oligonucleotide pool. In

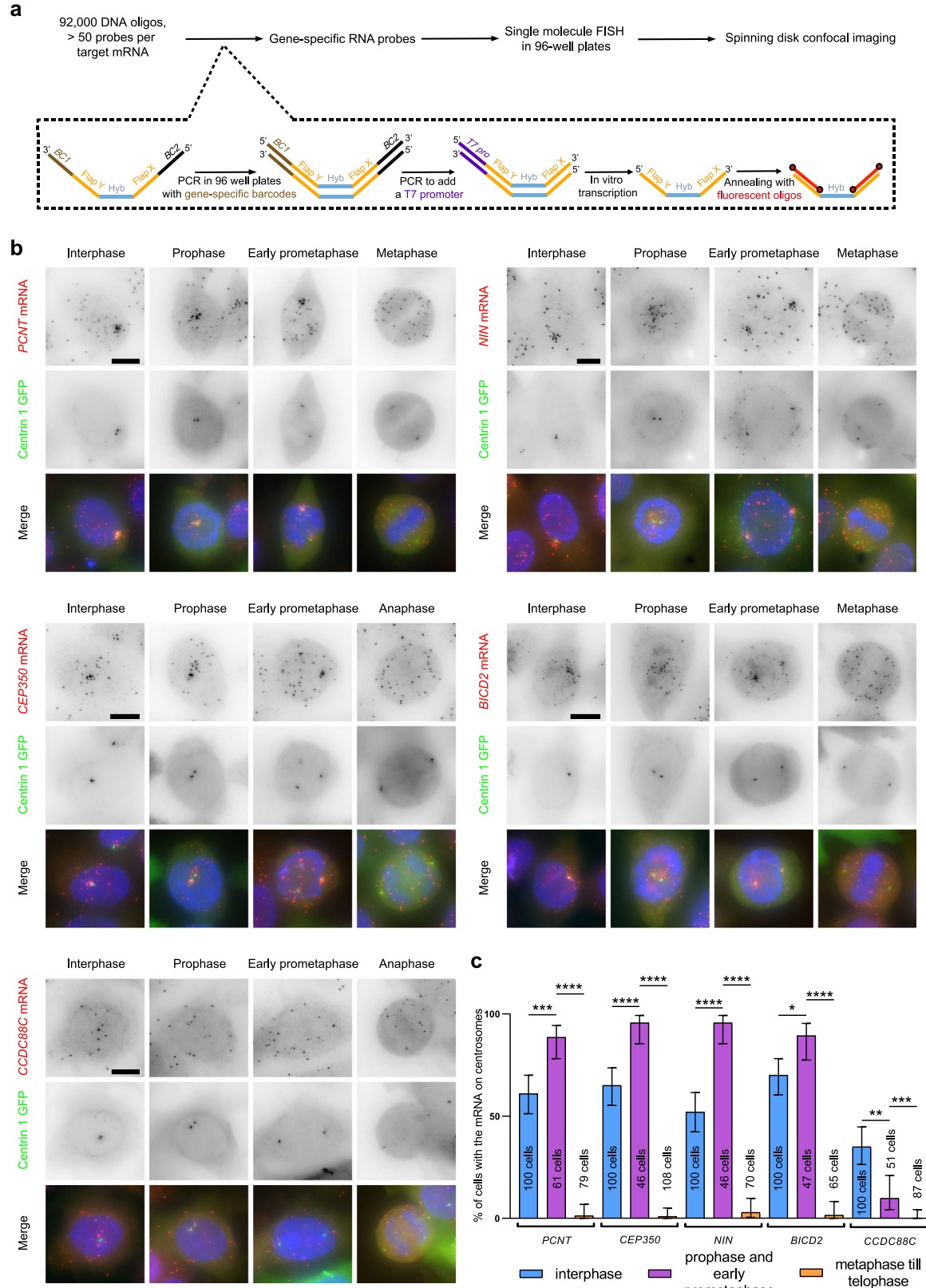

addition, the probes can be used individually or combined in different colors, allowing a flexible experimental design.

We screened a total of 602 genes using HeLa cells stably expressing a Centrin1–GFP fusion to label centrosomes. High-throughput spinning-disk microscopy was used to acquire full 3D images at high resolution (200 nm lateral and 600 nm axial), and

two sets of images were recorded, to image either interphase or mitotic cells (see "Materials and methods"). Centrosomal mRNA enrichment was assessed by manual annotations of the images. These analyses yielded several localized mRNAs, including six that concentrated near centrosomes (Table 1 and Table S1). The localization of these mRNAs was then confirmed by performing

**Fig. 1 An mRNA screen identifies new mRNAs localizing to centrosomes. a** Summary of the high-throughput smFISH pipeline. Top: starting from a pool of DNA oligonucleotides, gene-specific RNA probes are generated. SmFISH is performed on cells grown in 96-well plates, which are imaged by spinning disk confocal microscopy. Bottom: schematic representation of probe generation. BC: gene-specific barcode; FLAP X and Y: shared overhang sequences that hybridize with TYE-563-labeled LNA oligonucleotides (shown in red); Hyb: hybridization sequence-specific of the target mRNA, T7 pro: T7 RNA polymerase promoter. **b** Micrographs of HeLa cells stably expressing Centrin1–GFP and imaged by wide-field microscopy in interphase and mitosis. Left and red: Cy3 fluorescent signals corresponding to *PCNT, CEP350, BICD2, NIN,* or *CCDC88C* mRNAs labeled by low-throughput smiFISH; middle and green: fluorescent signals corresponding to the Centrin1–GFP protein. Blue: DNA stained with DAPI. Scale bars: 10 microns. **c** Bar graph depicting the percentage of cells in each phase, showing the centrosomal localization of each mRNA. Data were analyzed from the total number of cells indicated in the bars from three independent experiments and expressed as a percentage of cells with localized mRNA. Binomial proportion 95% confidence intervals are shown in each case and were calculated using the Wilson/Brown method. Statistical significance was evaluated using a two-sided Fisher's exact test. **** indicates a *p* value of <0.0001, *** a *p* value of around 0.0001, ** a *p* value of <0.01, * a *p* value of <0.05, and ns means nonsignificant.

low-throughput smiFISH[34]. The results confirmed that the six-candidate mRNAs accumulated at centrosomes during interphase and/or mitosis. These transcripts included PCNT and *NUMA1* mRNA that was also recently identified by us and others[8,24], as well as several new ones: *NIN, BICD2, CCDC88C,* and *CEP350* (Fig. 1b, Table 1). Taking into account *ASPM* and *HMMR* that were also recently identified[8,24], a total of eight mRNAs thus localize at centrosomes in human cells. These transcripts encode proteins that regulate various aspects of centrosome maturation, spindle positioning, and MT dynamics. Interestingly, the localization of these mRNAs varied during the cell cycle. *CCDC88C* mRNA localized during interphase but not mitosis. *PCNT, NIN, BICD2, HMMR,* and *CEP350* mRNAs localized during interphase and early mitosis but delocalized at later mitotic stages (Fig. 1c, and see below). In contrast, *NUMA1* and *ASPM* mRNAs only localized during mitosis (see below). Since all these mRNAs codes for centrosomal proteins, centrosomes thus appear to have a dedicated translational program that is regulated during the cell cycle.

**ASPM, NUMA1, and HMMR proteins localize to centrosome at specific cell cycle stages.** We then focused on *ASPM, NUMA1,* and *HMMR,* and analyzed in more detail their expression and localization. We first analyzed the expression of their respective proteins during the cell cycle. For this, we took advantage of HeLa Kyoto cell lines that stably express bacterial artificial chromosomes (BAC) containing the entire genomic sequences of the genes of interest, and carrying a C-terminal GFP tag[8,35]. These BACs contain all the gene regulatory sequences and are expressed at near-endogenous levels with the proper spatio-temporal pattern[35]. Time-lapse microscopy of single cells revealed that ASPM-GFP and HMMR-GFP expression rose progressively during interphase to culminate just before mitosis, while that of NUMA1-GFP appeared constant during the cell cycle. Interestingly, ASPM-GFP and NUMA1-GFP had similar localization patterns. Both proteins were mainly nucleoplasmic during interphase and precisely initiated centrosomal localization in prophase. During cell division, they accumulated at the spindle pole with weak labeling of the proximal spindle fibers (Supplementary Fig. 1a, b). In contrast, HMMR-GFP accumulated on the entire spindle throughout mitosis and furthermore concentrated on cytokinetic bridges in telophase. During interphase, it labeled MTs and localized to centrosomes several hours before cell division (Supplementary Fig. 1c).

**ASPM, NUMA1, and HMMR mRNAs localize to centrosome at the same time as their proteins.** We next determined the localization of *ASPM, NUMA1,* and *HMMR* mRNAs during the different mitotic phases. We again used the GFP-tagged BAC HeLa cell lines to correlate protein and mRNA localization. SmFISH was performed against the GFP RNA sequence using a set of 44 Cy3 labeled oligonucleotide probes[8]. In interphase,

*ASPM-GFP* and *NUMA1-GFP* mRNAs and proteins did not localize to centrosomes as previously reported[8], while *HMMR-GFP* mRNAs and its protein co-localized to centrosomes in a fraction of the cells. During mitosis, ASPM-GFP mRNAs and protein were enriched together on mitotic centrosomes across all phases of cell division (Fig. 2a, b). In contrast, *NUMA1* and *HMMR* mRNAs only accumulated at centrosomes during the early stages of cell division, prophase, and prometaphase, where they co-localized with their protein. A random mRNA distribution was seen during metaphase and anaphase, although both proteins still remained on the mitotic spindle (Fig. 2c, d). Unlike *NUMA1,* the centrosomal localization of *HMMR* mRNA and protein was re-established in telophase, where they accumulated together at the cytokinetic bridges (Fig. 2e, f).

To detail these findings, we performed two-color smFISH experiments detecting one BAC-GFP mRNA in Cy3 and an endogenous mRNA in Cy5. We analyzed all pairwise combinations of *ASPM, NUMA1,* and *HMMR* mRNAs. To gain more precision, we divided each of prophase, prometaphase, and telophase into two sub-phases, early and late (see "Materials and methods"). During the early prophase, *NUMA1* and *HMMR* mRNAs could be seen on centrosomes but not *ASPM* mRNAs that only joined during the late prophase (Supplementary Fig. 2). During early prometaphase, all three mRNAs were enriched on centrosomes. However, the centrosomal localization of *NUMA1* and *HMMR* mRNAs became much less frequent starting at late prometaphase, while that of *ASPM* mRNA could still be observed in metaphase and anaphase (Supplementary Figs. 2 and 3). Finally, *ASPM* but not *HMMR* mRNAs accumulated on centrosomes during early telophase, while the opposite was observed at late telophase. Interestingly, the three mRNAs never perfectly co-localized on centrosomes at any of the cell-cycle stages: certain peri-centrosomal regions were occupied by one transcript while others contained the other mRNA (Supplementary Figs. 2 and 3). Taken together, these data demonstrate a fine-tuning of spatio-temporal dynamics for the centrosomal localization of *ASPM, NUMA1,* and *HMMR* mRNAs, with each mRNA localizing at a specific stage and place during cell division.

**The localization of the eight centrosomal mRNAs is inhibited by puromycin but not cycloheximide.** Next, we analyzed the localization mechanism of these mRNAs and first questioned whether localization requires translation. To this end, we used a HeLa cell line expressing Centrin1–GFP to label centrosomes and treated it for 20 min with either cycloheximide, which blocks ribosome elongation, or puromycin, which induces premature chain termination. We first analyzed the mRNAs localizing during interphase (*NIN, BICD2, CCDC88C, CEP350, HMMR,* and *PCNT*). Remarkably, these six mRNAs became delocalized after puromycin treatment while cycloheximide had no effect (Fig. 3a, b). Long puromycin treatments prevent entry into mitosis. However, a 5-min incubation was sufficient to inhibit

**Table 1 Summary of centrosomal mRNAs in humans and their *Drosophila* orthologs.**

| Human gene | RNA localization | Protein localization | *Drosophila* orthologs | RNA localization | Protein localization |
|---|---|---|---|---|---|
| CCDC88C | Centrosome (this study) | Centrosome, nucleus | Girdin | Centrosome and cell division apparatus[31,32] (this study) | Centrosome, cytoplasm |
| NIN | Centrosome (this study) | Centrosome, nucleus | Bsg25D | Centrosome[31,32] (this study) | Centrosome |
| BICD2 | Centrosome (this study) | Centrosome, nuclear pore, nuclear envelope, Golgi, plasma membrane | BICD | Centrosomal (this study), cell division apparatus[31,32] | Centrosome, microtubules |
| CEP350 | Centrosome (this study) | Centrosome, mitotic spindle, nucleus | n/a | n/a | n/a |
| PCNT | Centrosome[24] | Centrosome, cytosol | Plp | Centrosome and cell division apparatus[31,32] | Centrosome |
| ASPM | Centrosome[8,24] | Mitotic centrosomes, mitotic spindle, nucleus | Asp | Centrosomal (this study), cell division apparatus[31,32] | Centrosomes, mitotic spindle poles, nucleus |
| NUMA1 | Centrosome[8] | Mitotic centrosomes, mitotic spindle, nucleus, plasma membrane | Mud | Centrosome (this study) | Centrosomes, mitotic spindle poles, nucleus |
| HMMR | Centrosome[8] | Centrosome, mitotic spindle, microtubules | n/a | n/a | n/a |
| n/a | n/a | n/a | Cen | Centrosome and cell division apparatus[31,32] | Centrosome, cytoskeleton, cytoplasm |
| CCP110 | Random (this study) | Centrosome, cilia basal body, cytoplasm | Cp110 | Centrosome and cell division apparatus[31,32] | Centrioles, cytoskeleton |

*n/a* not available

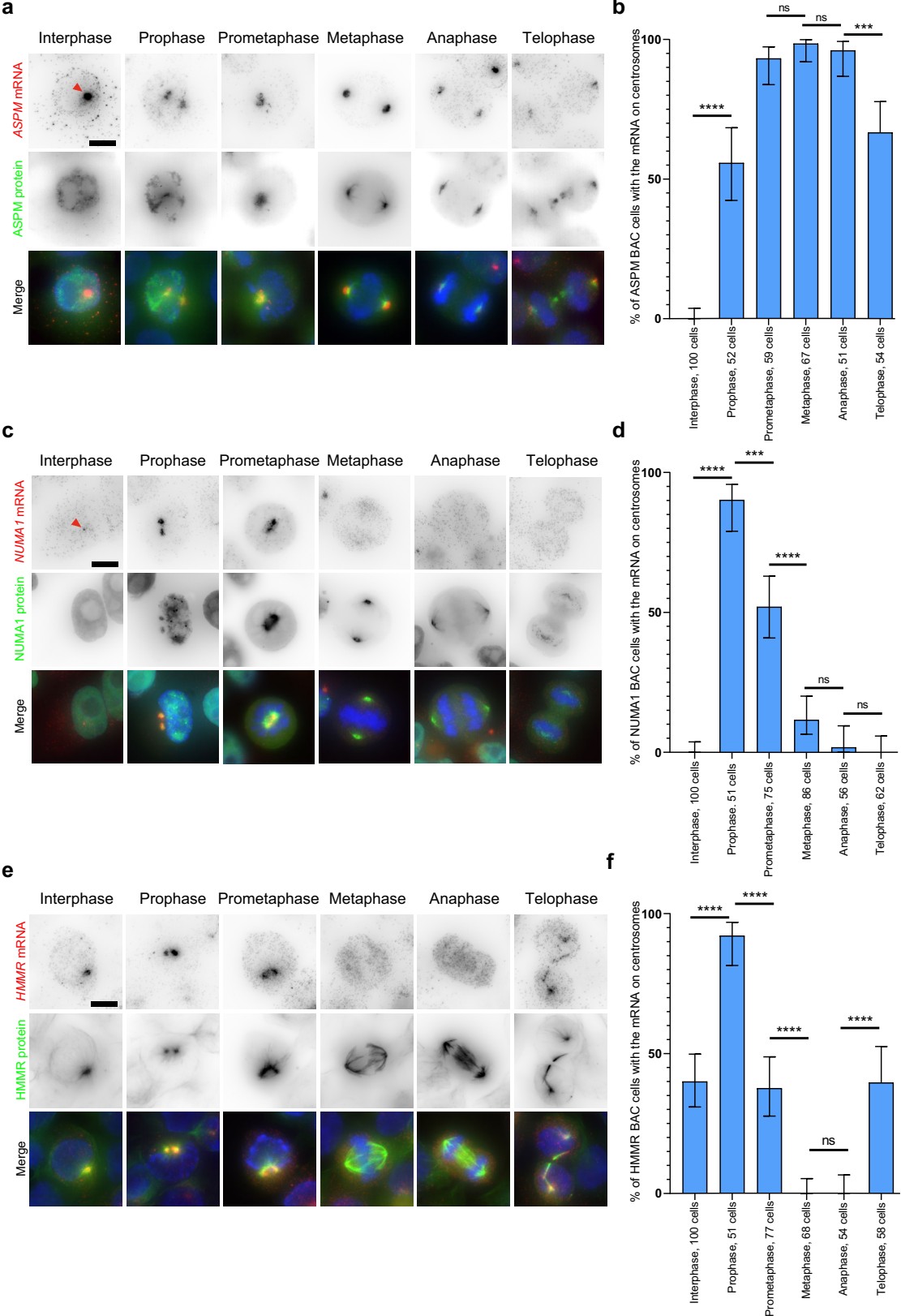

the centrosomal localization of *ASPM*, *NUMA1*, and *HMMR* mRNAs at all the mitotic phases in which they normally localize, while cycloheximide still had no effect (Supplementary Figs. 4–6). To validate these findings and add more quantitative depth, we performed an automated analysis on >40,000 cells

(Fig. 3c, d). SmFISH was performed in 96-well plates using HeLa–Centrin1–GFP cells and images were acquired in 3D with an automated spinning disk microscope equipped with a high-resolution objective (63× oil). Mitotic cells were sorted into the different mitosis phases with an automated classifier working

**Fig. 2 Differential centrosomal localization of *ASPM, NUMA1,* and *HMMR* mRNAs during mitosis. a** Micrographs of HeLa cells expressing an ASPM-GFP BAC and imaged at different phases of the cell cycle. Top and red: Cy3 fluorescent signals corresponding to *ASPM* mRNAs labeled by smFISH; middle and green: GFP signals corresponding to the ASPM protein. Blue: DNA stained with DAPI. Scale bar: 10 microns. The red arrow indicates a transcription site. **b** Bar graph depicting the percentage of cells in each phase, showing the centrosomal localization of *ASPM* mRNA. Data were analyzed from the total number of cells indicated in the bars from three independent experiments and expressed as a percentage of cells with localized mRNA. Binomial proportion 95% confidence intervals are shown in each case and were calculated using the Wilson/Brown method. Statistical significance was evaluated using a two-sided Fisher's exact test. **** indicates a *p* value of <0.0001., *** a *p* value of around 0.0001, and ns means nonsignificant. **c, d** Legend as in **a** and **b**, but for HeLa cells containing a NUMA1-GFP BAC. **e, f** Legend as in **a** and **b**, but for HeLa cells containing an HMMR-GFP BAC.

---

with 2D image projections using the DAPI signal. These cells were used to count the fraction of cells with localized mRNAs during mitosis (Supplementary Figs. 4–6). For all cells (mitotic and interphasic), centrosomes and single mRNAs were automatically detected on 2D image projections, and the proportion of mRNAs less than 2 mm away from a centrosome was computed in each cell. This was measured for thousands of cells in each condition, either without treatment or after a 10 min exposure to Puromycin or Cycloheximide. As compared to several control mRNAs (*TRIM59, TTBK2, KIF1C,* and *DYNC1H1*), *NIN, BICD2, CCDC88C, CEP350, PCNT, HMMR, NUMA1,* and *ASPM* accumulated to high levels near centrosomes, with a large cell-to-cell variability. Puromycin nearly abolished this localization while Cycloheximide had little effects. Since cycloheximide inhibits translation but leaves the nascent peptide chain on ribosomes, while puromycin removes it, our data suggest that mRNA localization to centrosomes requires the nascent peptide. RNA localization is thus expected to occur co-translationally for all the eight mRNAs, pointing toward a common localization mechanism.

**Translation of *ASPM* coding sequence is necessary and sufficient for localizing its mRNA at centrosomes.** To investigate how mRNAs localize to centrosomes in more detail, we focused on *ASPM* and first asked whether the 5′ and 3′ UTRs were necessary for its localization. To this end, a full-length *ASPM* mouse coding sequence (CDS) was fused to the C-terminus of GFP and expressed via transient transfection in HeLa Kyoto cells. To detect mRNAs produced from this reporter only, we performed smFISH with probes directed against the GFP RNA sequence. Mitotic cells expressing the plasmid could be identified by the accumulation of GFP-ASPM, which localized on centrosomes and the mitotic spindle. Interestingly, we could detect ASPM-GFP mRNAs on mitotic centrosomes in most of the transfected cells (Fig. 4a). We then tested whether the 3′ UTR of *ASPM* has some localization activity. We transfected a construct carrying the *Firefly luciferase* (*FFL*) CDS fused to the 3′ UTR of *ASPM* and performed smFISH against the luciferase sequence. This revealed that *FFL* CDS-ASPM 3′ UTR single molecules did not accumulate on centrosomes labeled by anti-g-tubulin immunofluorescence (IF; Fig. 4a). As an additional control, we performed smFISH against the GFP sequence in the HeLa cell line expressing Centrin 1–GFP. While the Centrin 1–GFP protein accumulated at the centrosome, its mRNA was distributed throughout the cytoplasm, indicating that adding a GFP tag did not promote centrosomal mRNA localization. Taken together, these constructs demonstrated that the 5′ and 3′ UTRs of *ASPM* mRNA are not required for its centrosomal enrichment.

Next, we explored how the same *GFP-ASPM* CDS mRNA would localize if the nascent ASPM protein was not translated. To test this, we introduced a stop codon between the GFP and *ASPM* CDS, generating a *GFP-stop-ASPM* construct. Transient transfection showed a diffuse GFP signal as expected. Interestingly, mRNAs translating this reporter failed to localize to mitotic centrosomes labeled by an IF against gamma-tubulin (Fig. 4b, c).

Taken together, this demonstrated that the nascent ASPM polypeptide is required for trafficking its own transcript to mitotic centrosomes.

**ASPM mRNAs are actively transported toward centrosomes and anchored on the mitotic spindle.** To gain more insights into the localization mechanism, we imaged the endogenous *ASPM* mRNAs in living mitotic cells. To this end, we inserted 24 MS2 repeats in the 3′ UTR of the endogenous gene, using CRISPR/Cas9-mediated homology-directed repair in HeLa Kyoto cells (Fig. 5a). Heterozygous clones were confirmed by genotyping (Supplementary Fig. 7a). Moreover, two-color smFISH performed with MS2 and *ASPM* probes showed that the tagged mRNA accumulated at centrosomes in mitosis (Fig. 5b), indicating that the MS2 sequences did not interfere with localization. We then stably expressed low levels of the MS2-coat protein (MCP) fused to GFP and a nuclear localization signal (MCP-GFP-NLS). This fusion protein binds the MS2 repeat and allows to visualize the tagged RNA in living cells[10]. Indeed, mitotic cells expressing MCP-GFP-NLS displayed diffraction-limited fluorescent spots that localized near the centrosomes. Moreover, these spots co-localized with single RNA molecules revealed with probes against either endogenous *ASPM* mRNA or the MS2 tag, indicating that binding of MCP-GFP-NLS to the tagged mRNA did not abolish RNA localization to the centrosome (Supplementary Fig. 7b).

Next, we performed live-cell experiments. We labeled DNA using SiR-DNA to identify the mitotic phase and we imaged cells in 3D at a rate of 2–4 fps using spinning disk microscopy. During prometaphase, three populations of *ASPM* mRNA molecules were observed: (i) mRNAs diffusing in the cytosolic space, (ii) immobile molecules that corresponded to mRNAs localizing at the centrosome, and (iii) mRNAs undergoing directed movements toward the centrosome (Fig. 5c). The thickness of mitotic cells yielded low signal-to-noise ratios which, combined with the rapid movements of the mRNAs, made single-particle tracking with automated software difficult. We thus manually tracked mRNA molecules undergoing directed movements and observed that they moved at speeds ranging from 0.5 to 1 μm/s (Fig. 5e), which is compatible with motor-mediated transport[36].

In metaphase and anaphase where the mRNA accumulates at centrosomes, we expected several possibilities for the movement of mRNA molecules: (i) stable anchoring to the centrosome; (ii) diffusion within a confined space around the centrosome; or (iii) diffusion away from centrosomes and re-localization by a motor-dependent mechanism. Live imaging revealed that *ASPM* mRNA localizing at the mitotic centrosome did not diffuse and were immobile (Fig. 5d; Supplementary Movie 1). In addition, directed movements toward centrosomes were also observed in mid-mitosis albeit to a lesser extent than in prometaphase (Fig. 5d arrows; Supplementary Movie 1). Interestingly, we also observed that some *ASPM* mRNAs were attached to the spindle fibers rather than on the spindle poles (Supplementary Movie 2). Taken together, these live imaging experiments demonstrated that *ASPM* mRNAs are actively transported to the mitotic

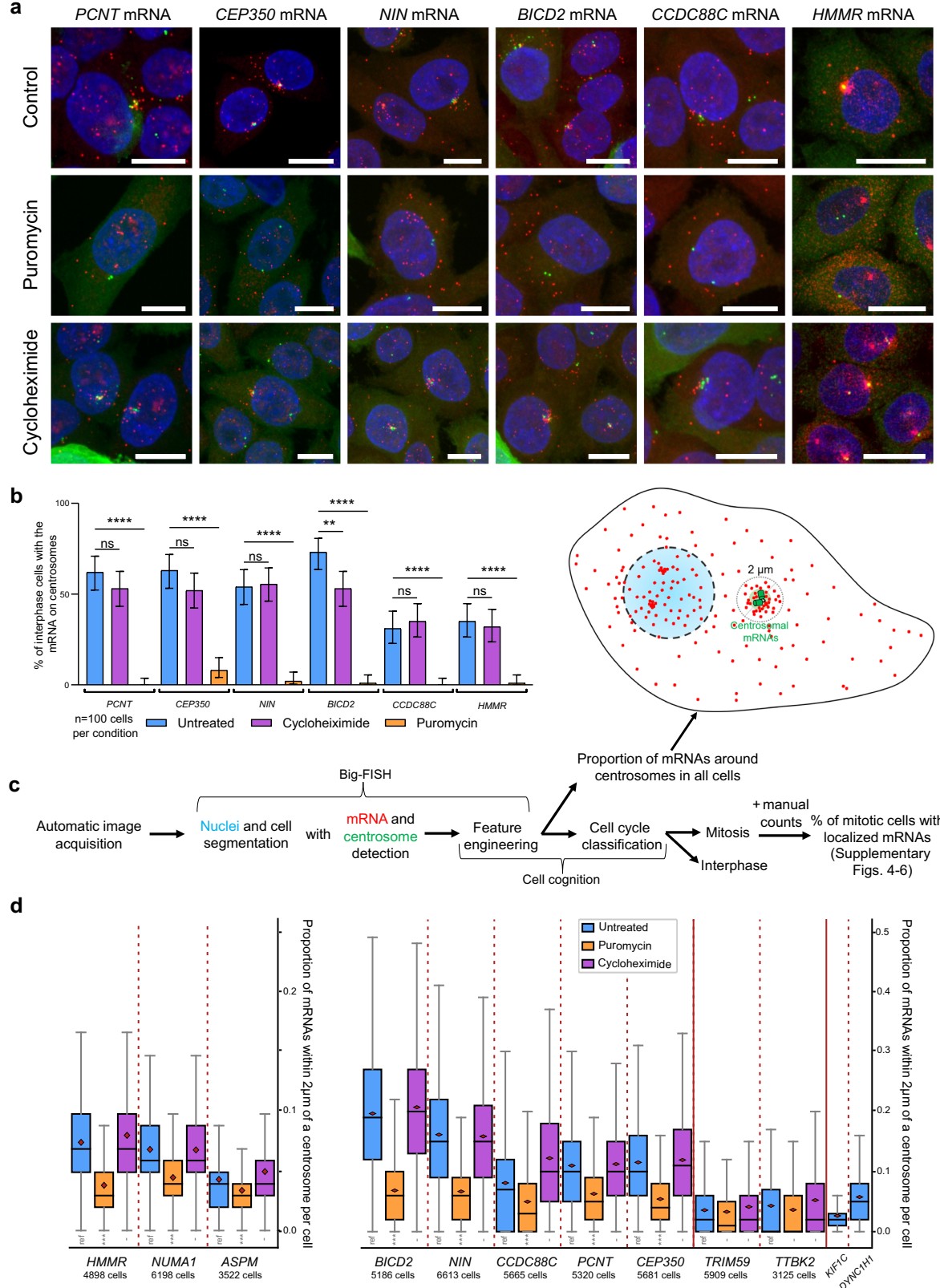

centrosomes at the onset of mitosis and are then anchored on the spindle poles and fibers.

**ASPM polysomes are actively transported toward centrosomes during prophase and prometaphase.** Since the localization of

ASPM transcripts required translation in *cis* and might thus involve the transport of translated mRNAs, we next imaged ASPM polysomes. To this end, we used the SunTag system that allows to image nascent polypeptide chains[7,37–40]. The SunTag is composed of a repeated epitope inserted in the protein of interest

**Fig. 3 Translation initiation is required for the localization of the centrosomal mRNAs. a** Micrographs of HeLa cells expressing either Centrin1–GFP or an HMMR-GFP BAC, and treated with cycloheximide or puromycin acquired using an automated spinning disk confocal microscope. Red: Cy3 fluorescent signals corresponding to *PCNT, CEP350, BICD2, NIN, CCDC88C,* or *HMMR-GFP* mRNAs labeled by smiFISH or smFISH; Green: either GFP signals corresponding to Centrin1–GFP or Cy5 signals corresponding to anti-g-tubulin immunofluorescence for HMMR. Blue: DNA stained with DAPI. Scale bars: 10 microns. **b** Bar graph depicting the percentage of interphase cells showing centrosomal mRNA localization after either a puromycin or cycloheximide treatment, as well as control cells. Data were analyzed from 100 cells per condition counted from two independent experiments and expressed as a percentage of cells with localized mRNA. Binomial proportion 95% confidence intervals are shown in each case and were calculated using the Wilson/Brown method. Statistical significance was evaluated using a two-sided Fisher's exact test. **** indicates a *p* value of <0.0001., ** a *p* value of <0.01, and ns means nonsignificant. **c** Schematic of the automated analysis pipeline. **d** Box plots depicting the proportion of centrosomal mRNAs in cells (interphase and mitotic), for the indicated mRNAs and cell treatments. Values were computed for each cell and the number of cells analyzed is indicated. The distribution of the values is depicted in the boxed plot. The colored area corresponds to the second and third quartiles, the mean to the black vertical bar, and the median to the red diamond. The whiskers equal 1.5 the interquartile range. The right graph represents endogenous centrosomal mRNAs, while the left one represents BAC-transcribed mRNAs. Significance was evaluated with a one-sided Welch's *t* test. ***: null hypothesis rejected with a 0.1% significance level, and -: null hypothesis not rejected.

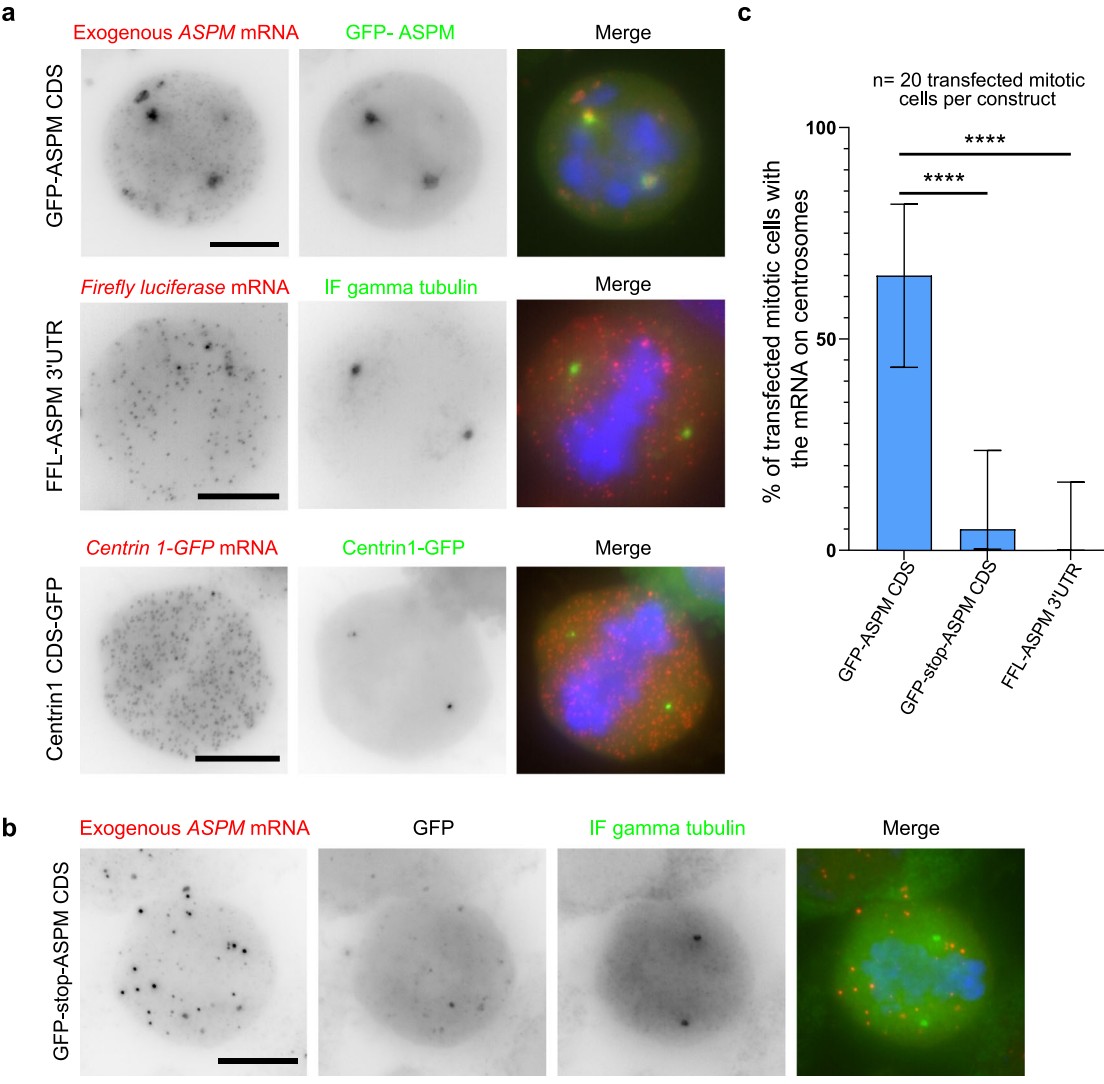

**Fig. 4 Translation of the *ASPM* CDS is necessary and sufficient for its centrosomal localization. a** Micrographs of mitotic HeLa cells either transiently expressing *GFP-ASPM* CDS, or *Firefly luciferase CDS-ASPM* 3'UTR; or stably expressing *centrin 1 CDS-GFP*. Left and red: Cy3 fluorescent signals corresponding to the exogenous mRNAs labeled by smFISH with probes against either the *GFP* or *Firefly luciferase* sequences; middle and green: GFP signals corresponding centrosomes labeled by the exogenous GFP-ASPM protein, gamma-tubulin immunofluorescence, or centrin1–GFP. Blue: DNA stained with DAPI. Scale bar: 10 microns. FFL *Firefly luciferase*. **b** Legend as in **a**, but with cells expressing *GFP-stop-ASPM* CDS and with an IF against gamma-tubulin. DNA is shown in blue, centrosomes in green and the mRNA in red. **c** Bar graph depicting the percentage of mitotic cells expressing a construct and showing the centrosomal localization of the exogenous mRNA. Data were analyzed from 20 mitotic cells per construct counted from two independent experiments and expressed as a percentage of cells with localized mRNA. Binomial proportion 95% confidence intervals are shown in each case and were calculated using the Wilson/Brown method. Statistical significance was evaluated using a two-sided Fisher's exact test. **** indicates a *p* value of <0.0001, and ns means nonsignificant.

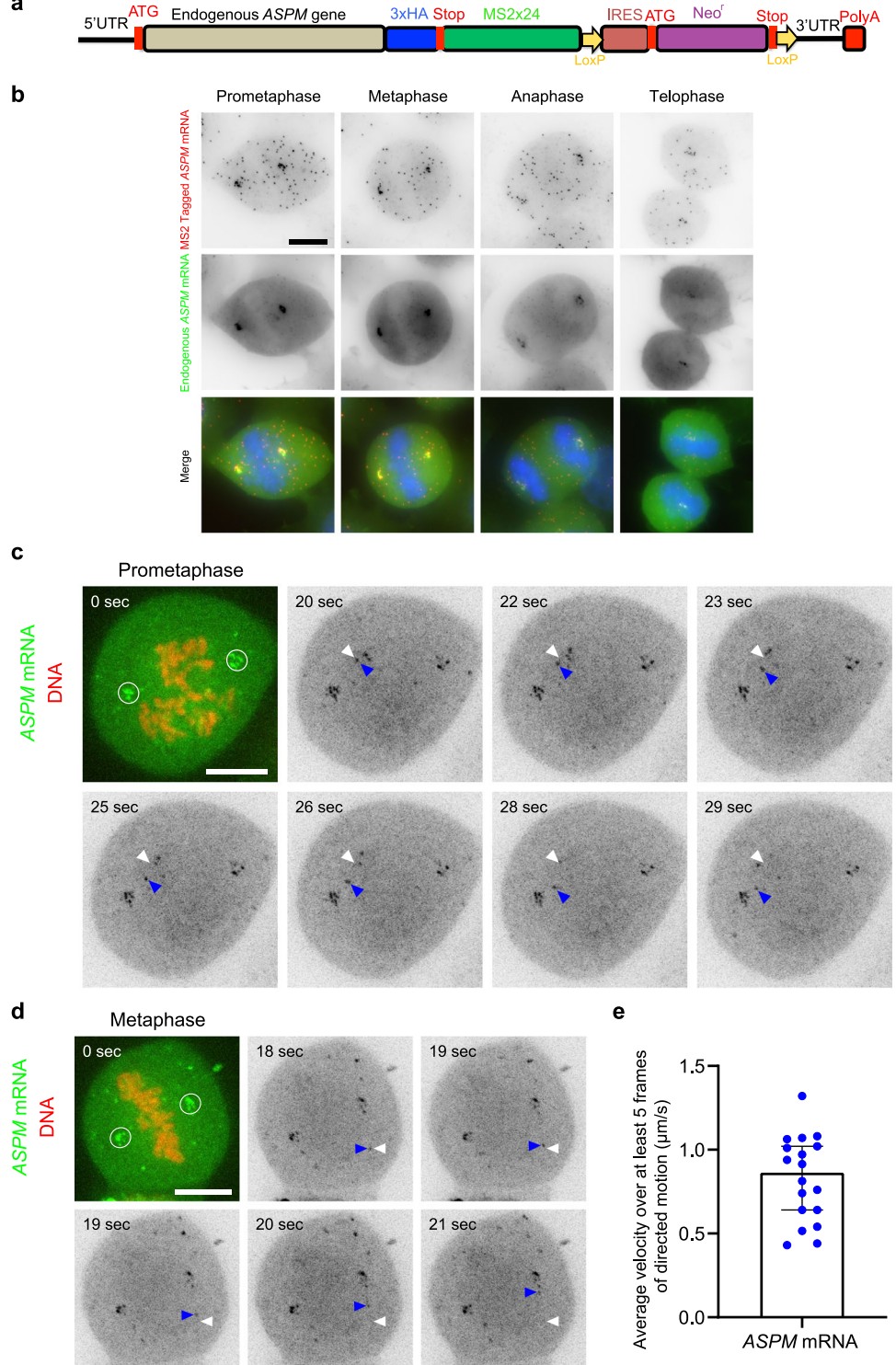

**Fig. 5 *ASPM* mRNAs undergo diffusion, directed movements, and anchoring to spindle poles and fibers across mitosis. a** Schematic representation showing the insertion of 24 MS2 repeats at the end of the *ASPM* gene using CRISPR-Cas9. HA human influenza hemagglutinin tag, Stop stop codon, IRES internal ribosome entry site, Neo[r] neomycin resistance, UTR untranslated region, PolyA poly A tail. **b** Images are micrographs of mitotic HeLa cells with an *ASPM-MS2x24* allele. Upper and red: Cy3 fluorescent signals corresponding to tagged mRNAs labeled by smFISH with MS2 probes; middle and green: Cy5 signals corresponding to the tagged and untagged *ASPM* mRNA detected by smiFISH with endogenous *ASPM* probes. Blue: DNA stained with DAPI. Scale bar: 10 microns. **c** Snapshots of *ASPM-MS2x24* cells expressing MCP-GFP-NLS and imaged live during prometaphase. In the first panel, the GFP signal is shown in green and corresponds to *ASPM* mRNAs labeled by the MS2-MCP-GFP-NLS. The Cy5 signal is shown in red and corresponds to DNA. Scale bar: 10 microns. Time is in seconds. The white circle displays *ASPM* mRNAs with limited diffusion. In subsequent panels, only the GFP signal is shown and is represented in black. White arrowheads indicate the starting position of an mRNA molecule, while blue ones follow its current position. **d** Legend as in **c** but for a cell captured in metaphase. **e** Bar graph showing the average speed of *ASPM* mRNA molecules displaying directed motion, calculated over at least five frames (or 2.5 s). The bar represents the median and error bars represent a 95% confidence interval. Data obtained across three independent experiments.

and a single-chain antibody fused to GFP. The binding of the fluorescent antibody to the epitope occurs when it emerges from the ribosome, and this allows visualizing nascent protein chains and polysomes in live cells. We engineered a HeLa Kyoto cell line with 32 SunTag repeats fused to the 5′ end of the *ASPM* gene, using CRISPR/Cas9-mediated homology-directed repair (Fig. 6a). Heterozygous clones were confirmed by genotyping (Supplementary Fig. 8a). The cells were then transduced with a lentivirus expressing the scFv mono chain antibody fused to GFP (scFv-sfGFP). Bright GFP foci were observed and confirmed to be polysomes based on both their sensitivity to puromycin (Supplementary Fig. 8b) and co-localization with endogenous *ASPM* mRNA by smiFISH (Fig. 6b). Moreover, *SunTag-ASPM* mRNAs accumulated on mitotic centrosomes (Supplementary Fig. 8c), indicating that the tagging process did not abolish centrosomal RNA targeting.

We first imaged SunTag-ASPM polysomes and mRNA together in fixed cells (Fig. 6b). In the early prophase, most mRNAs and polysomes did not localize to centrosomes, in agreement with the smFISH data (Supplementary Fig. 2). In prometaphase, metaphase, and anaphase, the accumulation of the SunTag-ASPM mature protein at the spindle poles prevented visualizing ASPM polysomes at this location. However, some ASPM polysomes were observed outside the spindle area indicating that translation was pursued during the entirety of mitosis (Fig. 6b). At the end of telophase, *ASPM* mRNAs could be seen translated on the nuclear envelope as previously reported for cells in interphase[8].

We then performed live imaging in 3D at acquisition rates of 1–1.3 stacks per second, using spinning disk microscopy. We first imaged cells in prophase (Supplementary Movie 3). Remarkably, while many ASPM polysomes were dispersed in the cytoplasm at the beginning of prophase, they displayed rapid directed motions toward the centrosome, leading to their accumulation at this location at the end of the movie (Fig. 6c; Supplementary Movie 3). Single-particle tracking showed that an average of 51% of ASPM polysomes displayed such directed movements in the early prophase (Fig. 6d, e). Directed movements were also detected in prometaphase but less frequently (Supplementary Movie 4). Calculating the average velocity of polysomes undergoing directed movements showed that their speed ranged from around 0.25–1.5 μm/s, which is compatible with the velocities of both *ASPM* mRNAs and motor-dependent transport (Fig. 6f). Taken together, this data directly proved that ASPM polysomes were actively transported to the centrosome at the onset of mitosis.

**ASPM mRNAs are translated on MTs during interphase.** Similar to MS2-tagged *ASPM* mRNAs, some SunTag-ASPM polysomes did not co-localize with the spindle poles but rather with spindle fibers that were weakly labeled by the mature SunTag-ASPM protein (Fig. 6b; arrows). This prompted us to investigate in more details the role of MTs in the metabolism of *ASPM* mRNAs. We labeled MTs in living cells using a far-red dye (SiR-Tubulin) and performed sequential two-color 3D live imaging using a spinning disk microscope. We first imaged interphase cells and remarkably, we observed that many ASPM polysomes remained stably anchored to MTs during the course of the movies (66%; Fig. 7a–c; Supplementary Fig. 9a; Supplementary Movie 5). In addition, we also observed directed motion of single polysomes, albeit at a low frequency (around 4%). We then characterized in more details the movements of ASPM polysomes and for this we classified them in four categories: (i) polysomes localizing on MTs; (ii) localizing at the nuclear envelope (as previously reported[8]); (iii) neither localizing on MTs nor at the nuclear envelope, and thus free in the cytosol; and (iv) showing

directed transport (Fig. 7c; Supplementary Fig. 9a). The histogram of displacements between consecutive frames revealed a diffusion coefficient of 0.011 $\mu m^2/s$ for polysomes on MTs, 0.004 $\mu m^2/s$ for the ones on the nuclear envelope, and 0.041 $\mu m^2/s$ for those not on MT or the envelope (i.e., freely diffusing; Supplementary Fig. 9b–d). We also calculated the mean square displacement (MSD) as a function of time (Fig. 7d). This confirmed that the ASPM polysomes that were free in the cytosol diffused several folds faster than those bound to MTs or the nuclear envelope. As a control, we depolymerized MTs with a 10-min nocodazole treatment before starting imaging. SiR–tubulin labeling confirmed the absence of MTs in treated cells, and we then tracked ASPM polysomes, excluding the ones attached to the nuclear envelope (Supplementary Fig. 10a–c; Supplementary Movie 6). The histogram of displacements and MSD curve revealed a single population with a diffusion coefficient of 0.035 $\mu m^2/s$, similar to polysomes, not on MT or the nuclear envelope in untreated cells (Fig. 7d; Supplementary Fig. 9e). Overall, this showed that a large fraction of *ASPM* mRNAs are locally translated on MTs during interphase and that these polysomes are stably anchored to MTs, thereby limiting their diffusion.

**ASPM polysomes are transported to mitotic centrosomes by either sliding on MTs or being pulled with entire MTs.** To assess whether MTs are necessary for *ASPM* mRNA localization to centrosomes, we combined a brief nocodazole treatment (10 min) with smFISH, using the *SunTag-ASPM* clone. A 10-min treatment depolymerized MTs whereas centrosomes were still visible (Supplementary Fig. 10a; white arrowheads). *ASPM* mRNAs and polysomes no longer accumulated at mitotic centrosomes after MT depolymerization, despite the fact that the mRNAs were still translated (Supplementary Fig. 10d–f). Moreover, a cytochalasin D treatment that inhibits actin polymerization performed in HeLa Kyoto cells did not prevent endogenous *ASPM* mRNAs from accumulating on centrosomes (Supplementary Fig. 10g, h). This indicated that intact MTs, but not actin filaments, are required for *ASPM* mRNA localization.

We then performed dual-color imaging of MTs and ASPM polysomes during mitosis. Tracks were shorter than in the previous mono-color movies because maintaining high frame rates required the recording of only 3 Z planes in two-color experiments, as opposed to 15–20 in the single-color movies. Nevertheless, this allowed us to distinguish two types of movements toward centrosomes. In the first, ASPM polysomes rapidly slid along an immobile MT (Fig. 7e; Supplementary Movie 7). This likely corresponded to motor-driven movements of polysomes along with MT cables. In the second type of movements, an ASPM polysome is stably attached to a MT and both were pulled together toward the centrosome (Fig. 7f; Supplementary Movie 8). In this case, the MT appears to be hauled towards the centrosome and drags a tethered polysome with it. Indeed, MTs pulling and sliding has been previously described during mitosis[41,42]. This demonstrated that two types of movements exist to transport ASPM polysomes to mitotic centrosomes: sliding on MTs, and tethering to MTs coupled to MT remodeling.

**NUMA1 mRNAs and polysomes also display directed transport toward the centrosome in early mitosis.** To assess the generality of the mechanism found with *ASPM*, we tagged another centrosomal mRNA, *NUMA1*. Using CRISPR/Cas9-mediated homology-directed repair in HeLa Kyoto cells, we generated a clone with an MS2×24 tag in the *NUMA1* 3′UTR, and another clone with a SunTag fused to the N-terminus of the protein. Proper recombination was verified by genotyping (Supplementary

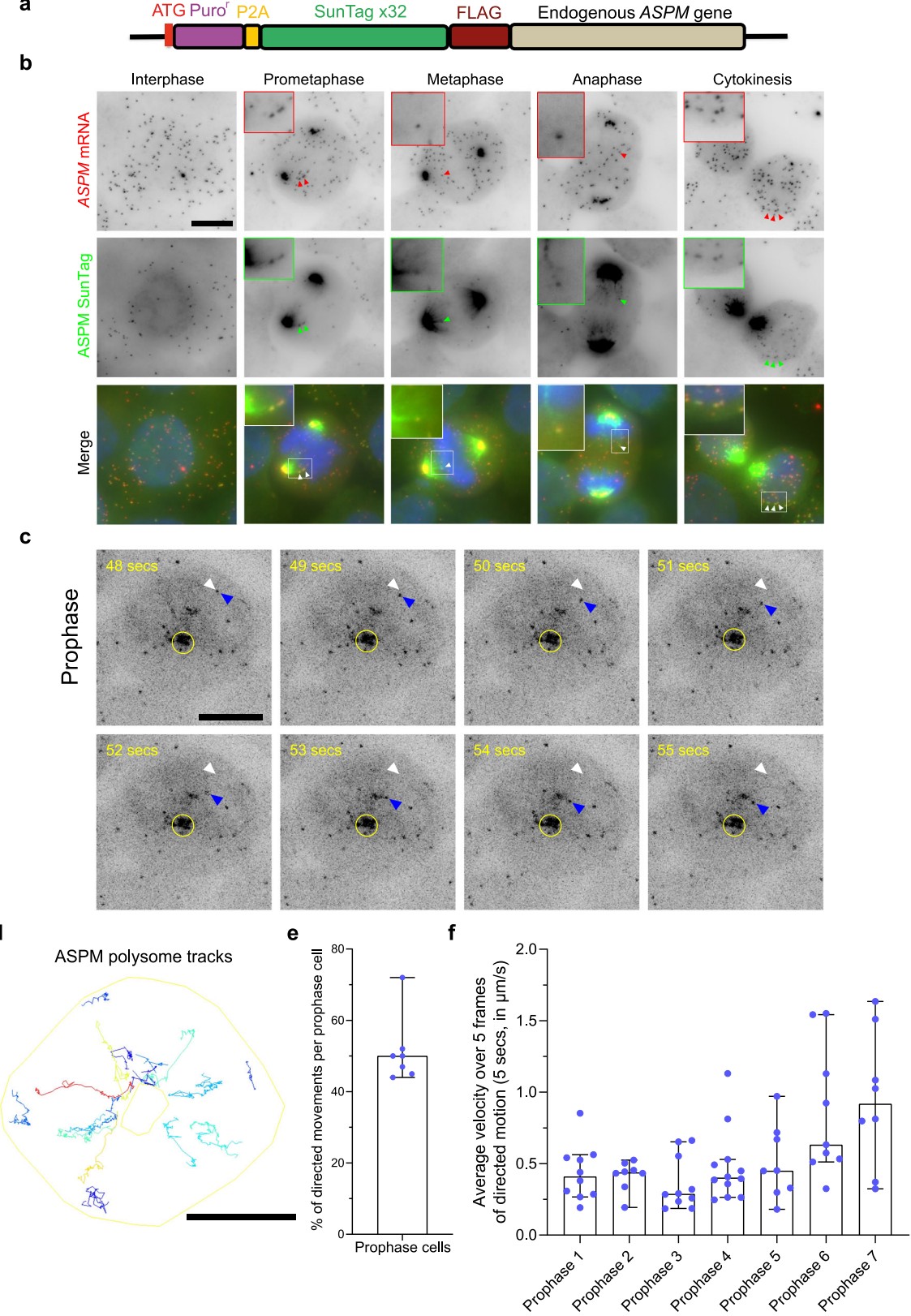

Fig. 11a, b). In the *NUMA1-MS2x24* clone, two-color smFISH performed against either the MS2 tag or the endogenous *NUMA1* mRNAs revealed that MS2 tagging did not prevent *NUMA1* mRNA from localizing to mitotic centrosomes (Fig. 8a). Likewise, the SunTagged *NUMA1* mRNA also localized to mitotic centrosomes (Supplementary Fig. 11c). Furthermore, in the SunTag

*NUMA1* clone, smiFISH against the endogenous *NUMA1* mRNA revealed that the mRNA colocalized with bright SunTag foci in both interphase and mitosis (Fig. 8b; Supplementary Fig. 11d). A puromycin treatment removed these bright cytoplasmic SunTag foci, confirming that they were NUMA1 polysomes (Supplementary Fig. 11e). Therefore, tagging the endogenous *NUMA1*

**Fig. 6 ASPM polysomes show directed movements toward the centrosome in early mitosis. a** Schematic representation of a cassette containing 32 SunTag repeats and with recombination arms to target insertion at the N-terminus of the endogenous *ASPM* gene. Puro[r] puromycin resistance, P2A self-cleaving signal, FLAG octapeptide FLAG tag. **b** Micrographs of a HeLa cell with a *SunTagx32-ASPM* allele and expressing scFv-sfGFP, imaged during interphase and mitosis. Upper and red: Cy3 fluorescent signals corresponding to tagged and untagged *ASPM* mRNAs labeled by smiFISH with probes against the endogenous *ASPM* mRNA; middle and green: GFP signals corresponding to ASPM polysomes and mature protein. Blue: DNA stained with DAPI. Scale bar: 10 microns. Red and green arrowheads indicate *ASPM* mRNAs and polysomes, respectively. White arrows indicate the overlay of red and green arrows. Insets represent zooms of the white-boxed areas. **c** Snapshots of HeLa cells with a *SunTagx32-ASPM* allele and expressing scFv-sfGFP, imaged live during prophase. The SunTag signal is shown in black and corresponds to ASPM polysomes and mature proteins. Scale bar: 10 microns. Time is in seconds. The yellow circle indicates ASPM mature protein. White arrowheads indicate the starting position of an ASPM polysome, while blue ones follow its position at the indicated time. **d** A TrackMate overlay of the same cell in (**c**) showing polysomes tracks. Color code represents displacement (dark blue lowest, red highest). The outer yellow outline represents the cell border, while the inner one represents centrosomes marked by the mature ASPM protein. Scale bar: 10 microns. **e** A bar graph showing the percentage of polysomes displaying directed movements toward the centrosome per prophase cells. Each dot indicates a cell and the bar represents the median. Error bars correspond to a 95% confidence interval. $N = 7$ cells examined over three independent experiments. **f** Bar graphs showing the average speed of ASPM polysomes displaying directed motions (calculated over at least five frames), in prophase cells. Each dot indicates a polysome trajectory. The bars represent the median and error bar a 95% confidence interval. $N = 7$ cells examined over three independent experiments.

mRNA with MS2 and SunTag repeats did not abolish its centrosomal localization.

Following stable expression of MCP-GFP-NLS or scFv-sfGFP, we imaged single *NUMA1* mRNAs and polysomes, respectively, in living interphase and mitotic cells. In interphase, we could observe some NUMA1 polysomes undergoing rapid rectilinear movements (Supplementary Fig. 11f). Remarkably, in prometaphase, we observed rapid directed motion of both mRNAs and polysomes toward the centrosome, indicating that NUMA1 polysomes are actively transported toward centrosomes at the onset of mitosis, similar to what we observed for ASPM polysomes (Fig. 8c, d; Supplementary Movies 9 and 10). We manually calculated the mean particle velocity. Both *NUMA1* mRNAs and polysomes had speeds of around 0.5–1 μm/s, which is compatible with motor directed movements and also similar to the velocities measured for *ASPM* (Fig. 8e). These live imaging experiments of endogenous transcripts and polysomes show that active polysome transport is a localization mechanism shared by several centrosomal mRNAs.

**Drosophila orthologs of the human centrosomal mRNAs also localize to centrosomes and also require the nascent protein.** We next examined whether centrosomal mRNA localization is evolutionary conserved. To this end, we investigated the localization of the *Drosophila* orthologs of the human centrosomal mRNAs. Out of the eight human mRNAs, five had clear orthologs: *ASPM* (*Asp*), *NUMA1* (*Mud*), *BICD2* (*BicD*), *CCDC88C* (*Girdin*), and *PCNT* (*Plp*). We used S2R+ cells as a model and co-labeled these mRNAs with centrosomes, using smFISH coupled to IF against gamma-tubulin. Remarkably, we observed centrosomal enrichment for four of these five *Drosophila* mRNAs (Fig. 9; Table 1). Moreover, while we could not obtain clear signals for the fifth mRNA in S2R+ cells (Plp), it was annotated as centrosomal in a previous large-scale screen (Table 1). Interestingly, we also observed a cell-cycle dependent localization for Asp, Mud, and Girdin mRNAs (Supplementary Fig. 12). A large fraction of cells in interphase did not localize Mud mRNAs whereas it localized to centrosomes during mitosis (metaphase, anaphase, and telophase; Fig. 9b; Supplementary Fig. 12). Asp mRNA showed a similar dynamic, with the exception that it was not localized in telophase. In contrast, Girdin mRNAs only accumulated on mitotic centrosomes during telophase (Supplementary Fig. 12). These observations indicate that a conserved cell-cycle regulated centrosomal translational program occurs in both human and *Drosophila* cells.

Finally, we tested whether mRNAs in *Drosophila* depend on their nascent peptide to localize to centrosomes. We performed

puromycin and cycloheximide treatments and co-labeled the mRNAs and centrosomes in S2R+ cells. As in human cells, puromycin treatment abolished centrosomal accumulation of all four mRNAs while cycloheximide did not (Fig. 9). This indicated that not only the identity of centrosomal mRNAs is conserved from human to *Drosophila*, but also the localization mechanism, which requires the nascent polypeptide in both cases.

## Discussion

Here, we studied centrosomal mRNA localization in human cells. We uncovered a complex choreography of mRNA trafficking at centrosomes, particularly during mitosis, and we provide definitive evidence for a nascent chain-dependent transport of polysomes by motors and MTs. Remarkably, both the identity of localized mRNAs and the localization mechanism appear conserved from *Drosophila* to humans.

**A targeted smFISH screen reveals that centrosomal mRNAs are conserved from human to *Drosophila*.** We used high-throughput smFISH to screen 602 genes encoding nearly the entire centrosomal proteome (see Material and methods), and we identified four new mRNAs localized at the centrosome (Table 1; Table S1). In total, eight human mRNAs now belong to this class: NIN, CEP350, PCNT, BICD2, CCDC88C, ASPM, NUMA1, and HMMR. All the corresponding proteins localize to the centrosome suggesting that their mRNAs are locally translated. Most of them also perform important centrosomal functions. NIN is localized to the sub-distal appendage of mother centrosomes and it functions in MT nucleation as well as centrosome maturation[43,44]. CEP350 is also localized to sub-distal appendages and it is important for centriole assembly and MT anchoring to centrosomes[45,46]. Pericentrin (PCNT) is a major component of the PCM and it plays a structural role by bridging the centrioles to the PCM[47]. BICD2 contributes to centrosomal positioning and to centrosomal separation at the onset of mitosis[48,49]. ASPM and NUMA1 are two MT minus-end binding proteins that accumulate at centrosomes during mitosis, and they control several aspects of spindle assembly and function[50,51]. Finally, HMMR acts to separate centrosomes and to nucleate MT during spindle assembly. It also modulates the cortical localization of NUMA-dynein complexes to correct mispositioned spindles[52]. The diversity of functions performed by these proteins suggests that RNA localization and local translation play an important role in the centrosome.

Remarkably, we found that five of the human centrosomal mRNAs had orthologs in *Drosophila*, and four of these localized

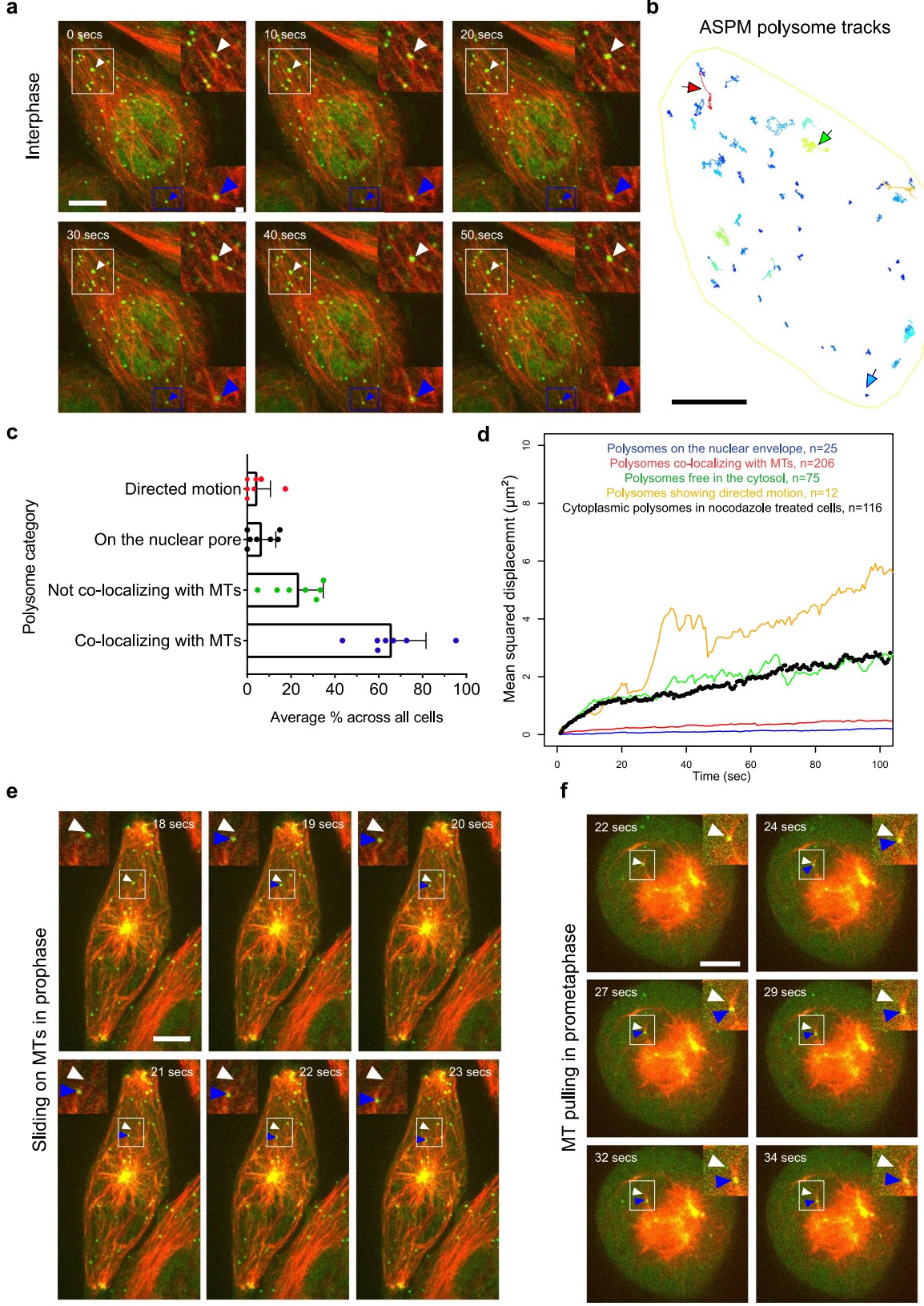

to centrosomes in S2R+ cells. Moreover, the fifth *Drosophila* mRNA, Plp, was reported to localize to centrosomes in *Drosophila* embryos (Table 1)[31,32]. These data show a striking and unprecedented degree of evolutionary conservation in RNA localization, where the same family of mRNAs is localized to the same subcellular site, from *Drosophila* to humans. This likely underlies conserved features in mRNA localization mechanism and/or function.

**A cell cycle-dependent translational program operates at centrosomes**. Our data reveal that centrosomal mRNA localization

**Fig. 7 ASPM polysomes are anchored to microtubules in interphase and transported toward the centrosome in early mitosis. a** Snapshots of *SunTagx32-ASPM* cells expressing scFv-sfGFP and imaged live during interphase with MT labeling. The SunTag signal is shown in green and corresponds to ASPM polysomes and mature proteins; the far-red signal is shown in red and corresponds to a tubulin staining. Scale bar: 10 microns. Time is in seconds. Upper and lower insets represent one-plane zooms of the white- and blue-boxed areas respectively. Arrowheads point to ASPM polysomes. **b** A TrackMate overlay of the same cell as in (**a**) showing the polysome tracks. Color code represents displacement (dark blue lowest, red highest). Scale bar: 10 microns. The red arrow indicates a track showing directed movement, the green one is a track that does not localize with MTs, and the blue one is a track that localizes with MTs. **c** The average percentage of ASPM tracks per cell grouped in four categories depending on subcellular localization and behavior in interphase. Data are represented as mean values and error bars represent the standard deviation. Data obtained across two independent experiments. Polysomes free in the cytosol: polysomes neither on MTs, nor on the nuclear envelope. **d** Graph showing the mean squared displacement (MSD, in $\mu m^2$) of polysomes as a function of time (in seconds) for each of the indicated polysome category and after nocodazole treatment. Polysomes free in the cytosol: polysomes not on MT and not on the nuclear envelope. **e** Snapshots of *SunTagx32-ASPM* cells expressing the scFv-sfGFP and imaged live during prophase with MT labeling. The SunTag signal is shown in green and corresponds to ASPM polysomes and mature proteins; the far-red signal is shown in red and corresponds to a tubulin staining. Scale bar: 10 microns. Time is in seconds. Insets represent one-plane zooms of the white-boxed areas. White and blue arrowheads follow the initial and current position of a polysome, respectively. **f** Same legend as in (**e**), but during prometaphase.

varies with phases of the cell cycle. Two mRNAs specifically localized in mitosis (*ASPM* and *NUMA1*), one in interphase but not mitosis (*CCDC88C*), and five in interphase and early mitosis (*HMMR*, *BICD2*, *CEP350*, *PCNT*, *NIN*). The phase where most mRNAs (seven out of eight) localize is thus prophase. Moreover, the two mitotic mRNAs localized with different kinetics: *ASPM* localized during all mitotic phases while *NUMA1* only during prophase and prometaphase. Finally, *HMMR* was the only transcript that localized at the cytokinetic bridge at the end of cell division, together with its protein. Interestingly, the *Drosophila* centrosomal mRNAs also localize in a cell cycle-dependent way, indicating that this feature is also conserved during evolution. This shows the variety, complexity and precision of centrosomal mRNA localization, as well as its potential role during the centrosome cycle. These data demonstrate the existence of a unique and conserved translational program at centrosomes, which is cell cycle-regulated.

It is interesting to speculate why these eight proteins and not others are locally translated. Since they function in centrosome/spindle maturation and that this occurs over short time periods, having optimal amounts at centrosomes at the right time point of the cell cycle is crucial. Interestingly, most of these proteins have relatively large sizes (more than 2000 aa, with the exception of HMMR and BICD2), and it would thus take some time to synthesize and transport them. Local translational regulation may thus provide an efficient and rapid method for targeting them to the centrosome when needed. This is likely important during prophase where most mRNAs localize because it is a relatively short phase and also the site of important changes in centrosome composition and function. This notion might be crucial for *ASPM* in the context of proliferative divisions of neuroepithelial (NE) stem cells. In NE cells, knocking down *ASPM* mRNA leads to a reduction in spindle pole ASPM protein levels which reduces self-renewing symmetric cell division of NE cells leading to microcephaly[53]. This is caused by mitotic orientation defects, as well as cell cycle deregulations[54]. It would be interesting to see if *ASPM* mRNA is locally translated on centrosomes in NE stem cells and whether this contributes to maintaining their proliferative divisions.

Another possibility is that mRNA accumulation at centrosomes plays a structural role. An emerging model is that phase separation helps the formation of centrosomes[55]. Since RNAs are often critical components of phase-separated condensates[56], their accumulation at centrosomes could contribute to their formation. Finally, a likely and not exclusive possibility is that these eight proteins need to be assembled co-translationally with their partners at the centrosome. Co-translational folding occurs for many proteins and this can be facilitated by the presence of a protein's partner. This may

further provide an elegant mechanism for RNA localization (see below).

**Centrosomal mRNA localization occurs by active transport of polysomes and requires the nascent protein.** For all the eight centrosomal transcripts studied here, premature ribosome termination delocalized the mRNAs while freezing the ribosome and the nascent protein chain on the mRNA had no effect. In the case of a GFP-tagged *ASPM* mRNA, we further observed that preventing translation of the nascent ASPM protein via a stop codon abolished centrosomal RNA localization, indicating that translation is required in *cis*. Most importantly, polysomes coding for ASPM and NUMA1 were actively transported to centrosomes at rates compatible with motor-driven transport (0.5–1.5 µm/s). Together, this shows that centrosomal mRNA localization relies on an active mechanism driven by the nascent peptide. While RNA localization is often conceptualized as an RNA-driven process that transports silenced mRNAs, our data contradict this dogma and show that for centrosomal mRNAs, polysome transport mediated by the nascent protein is the rule.

These observations suggest that the nascent polypeptide contains a localization signal that would drive the polysome toward centrosomes. Interestingly, we found that ASPM polysomes are actively transported via two mechanisms. The first is motorized transport whereby a polysome slides on MTs. The second involves the pulling of an entire MT with an ASPM polysome attached to it. The N-terminal part of ASPM contains domains that bind MTs, either directly or with katanin[50]. It is possible that once these domains are translated, they cause the entire ASPM polysome to attach to MTs. In agreement with this possibility, local translation of *ASPM* at MTs can also be seen during interphase. Interestingly, all eight localized mRNAs encode proteins that either bind MTs (ASPM, NUMA1, HMMR, CCDC88C, and CEP350) or contribute to MT anchoring (NIN, PCNT, and BICD2). Similarly, many of these proteins directly or indirectly bind dynein (BICD2, NIN, CCDC88C, PCNT, NUMA1, and HMMR)[57]. These properties could thus be part of the transport mechanism.

The paradigm for translation-dependent RNA localization is that of secreted proteins. In this case, translation initially leads to the synthesis of the signal peptide, which is recognized by SRP. This halts the ribosome until the entire complex docks on the SRP receptor on the ER, where translation resumes. It is thus tempting to envision a scenario where ribosomes translating centrosomal mRNAs enter a pause and only resume translation after reaching the centrosome. It has been shown that unfolded domains can halt ribosomes[58,59]. Moreover, in a recent case of co-translational assembly, the ribosome enters a pause at a specific location, which

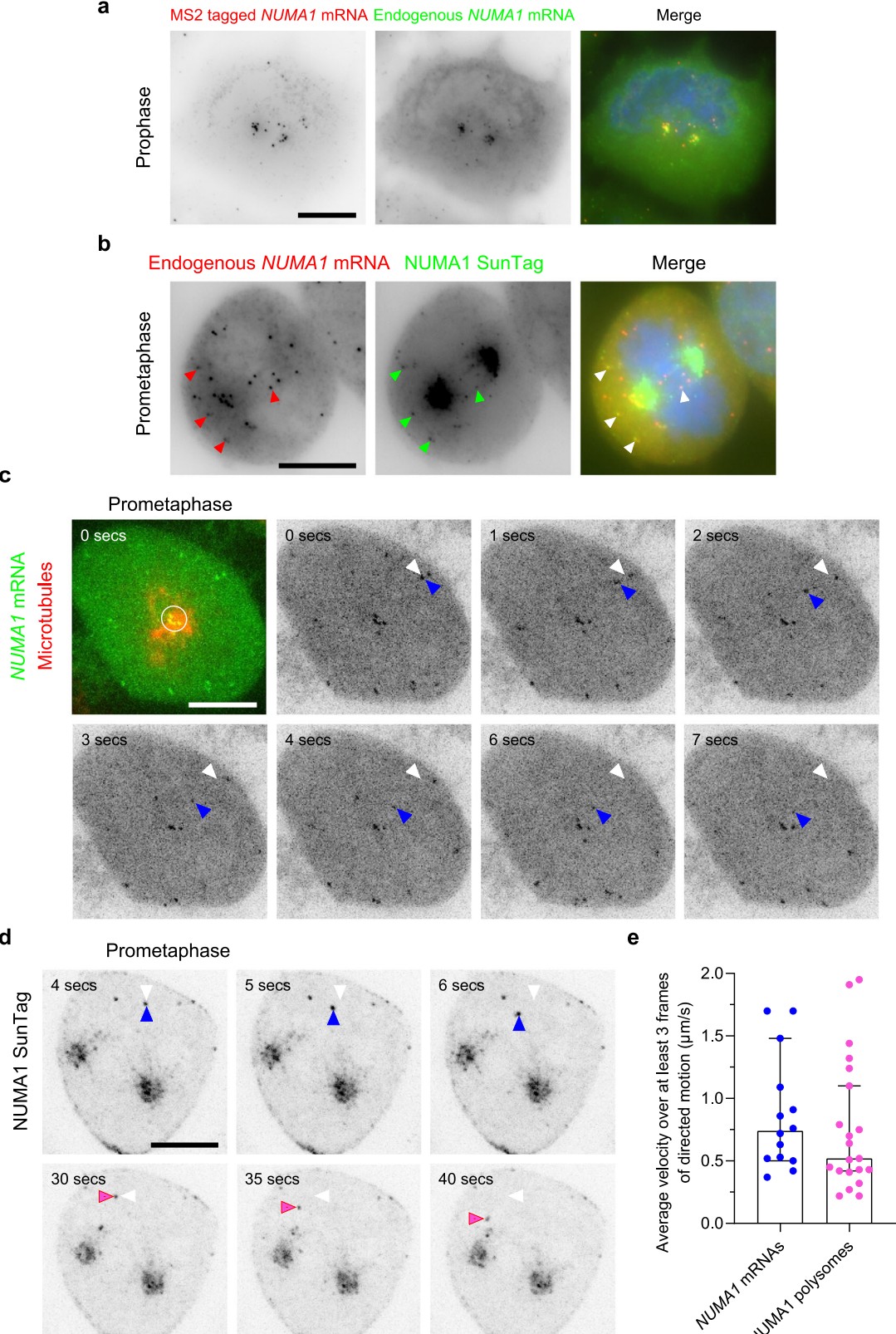

is relieved upon interaction with the partner of the nascent protein[60]. If indeed the proteins encoded by the centrosomal mRNAs need to be co-translationally assembled, it is possible that the domain responsible for this would remain unfolded before the polysome reaches the centrosome. It could thus halt the ribosomes while the nascent polypeptide located upstream of this unfolded domain could connect the polysome to transport systems and drive it to the centrosome. This could be an elegant and general mechanism that ensures RNA localization and local translation, as it could work at any place in the cell. This could explain why co-translational mRNA targeting appears to be a widespread mechanism in cell lines[8].

**Fig. 8 NUMA1 mRNAs and polysomes also show directed movements toward the centrosome in early mitosis. a** Images are micrographs of HeLa cells with a *NUMA1-MS2x24* allele and imaged in prophase. Left and red: Cy3 fluorescent signals corresponding to tagged mRNAs labeled by smFISH with MS2 probes; middle and green: Cy5 signals corresponding to the tagged and untagged *NUMA1* mRNA labeled by smiFISH with probes against the endogenous mRNA. Blue: DNA stained with DAPI. Scale bar: 10 microns. **b** Images are micrographs of *SunTagx32-NUMA1* cells expressing the scFv-sfGFP and imaged during prometaphase. Left and red: Cy3 fluorescent signals corresponding to *NUMA1* mRNAs labeled by smiFISH; middle and green: GFP signals corresponding to NUMA1 polysomes and mature protein. Blue: DNA stained with DAPI. Scale bar: 10 microns. Red and green arrowheads indicate *NUMA1* mRNAs and polysomes, respectively. White arrows indicate the overlay of red and green arrows. **c** Snapshots of *NUMA1-MS2x24* cells expressing MCP-GFP-NLS and imaged live during prometaphase. In the first panel, the GFP signal is shown in green and corresponds to *NUMA1* mRNAs labeled by the MS2-MCP-GFP-NLS. The far-red signal is shown in red and corresponds to microtubules. Scale bar: 10 microns. Time is in seconds. The white circle indicates *NUMA1* mRNAs with limited diffusion. In subsequent panels, only the GFP signal is shown and is represented in black. White arrowheads indicate the starting position of an mRNA molecule, while blue ones follow its current position. **d** Snapshots of *SunTagx32-NUMA1* cells expressing the scFv-sfGFP and imaged live during prometaphase. The SunTag signal is shown in black and corresponds to NUMA1 polysomes and mature proteins. Scale bar: 10 microns. Time is in seconds. White arrowheads indicate the starting position of two polysomes, while blue and pink ones follow their current position. **e** Bar graph showing the average speed of *NUMA1* mRNA and polysomes displaying directed motion calculated over at least three frames. The bar represents the median and error bars represent a 95% confidence interval. $N = 15$ cells for *NUMA1* mRNA and $N = 17$ cells for NUMA1 polysomes, both examined over three independent experiments.

## Methods

**Cell lines, culture conditions, and treatments**. HeLa-Kyoto cells and HeLa cells expressing or not Centrin1–GFP (a gift from Dr. B. Delaval) were grown in Dulbecco's modified Eagle's Medium (DMEM, Gibco) supplemented with 10% fetal bovine serum (FBS, Sigma-Aldrich), and 100 U/mL penicillin/streptomycin (Sigma-Aldrich). The collection of HeLa-Kyoto cell lines stably transfected with the GFP-tagged BACs was described previously[35], and these cells were grown in the same medium in addition to 400 μg/ml G418 (Gibco). All human cells were grown at 37 °C with 5% $CO_2$. S2R+ cells were grown in Schneider's Drosophila medium (Gibco) supplemented with 10% FBS (Sigma-Aldrich), and 100 U/mL penicillin/streptomycin (Sigma-Aldrich) at 25 °C. Drugs were used at the following final concentrations: 100 μg/ml for puromycin, 200 μg/ml for cycloheximide, 5 μg/ml for nocodazole, and 1 μg/ml for cytochalasin D. Treatment of cells with translation inhibitors was for 20 min (reduced to 5–10 min for mitotic cells when indicated). Treatment of cells with nocodazole and cytochalasin was for 10 and 30 min, respectively. Transfection of the *GFP+/− stop codon -ASPM CDS* and *FFL CDS-ASPM 3'UTR* constructs was done using JetPrime (Polyplus) and 2 μg of DNA were transfected overnight in a 6-well plate containing 2 ml of medium.

**Insertion of the MS2 cassette by CRISPR/Cas9**. The recombination cassettes contained 500 bases of homology arms flanking a 3×HA tag, a stop codon and 24 MS2 repeats. A start codon was placed after the MS2 repeats followed by an IRES, a neomycin resistance gene, and a stop codon. The IRES-Neo$^r$ segment was flanked by two LoxP sites having the same orientation. HeLa Kyoto cells were transfected using JetPrime (Polyplus) and a cocktail of four plasmids, including the recombination cassette and constructs expressing Cas9-nickase and two guide RNAs with an optimized scaffold[7]. Insertion was targeted at the stop codon of the *ASPM* and *NUMA1* genes. Cells were selected on 400 μg/ml G418 neomycin for a few weeks. Individual clones were then picked and analyzed by PCR genotyping, fluorescent microscopy, and smFISH/smiFISH with probes against both the endogenous *ASPM* or *NUMA1* mRNA and MS2 sequences. Stable MCP-GFP-NLS expression was then set up via retroviral infection. The sequences targeted by the guide RNAs were (PAM sequences are lowercase): TCTCTTCTCAAAACCCAATCtgg for *ASPM* guide 1, and GCAAGCTATTCAAATGGTGAtgg for *ASPM* guide 2; GAGGTCAGCATCGGGGACACAgg for *NUMA1* guide 1, and AGTGCCTTCTCTCAGCTCCCagg for *NUMA1* guide 2.

**Insertion of SunTag cassette by CRISPR/Cas9**. The recombination cassettes contained 500 bases of homology arms flanking a puromycin resistance gene translated from the endogenous ATG sequence, followed by a P2A sequence, 32 SunTag repeats, and a P2A-T2A-FLAG sequence in the case of *ASPM* or a P2A-FLAG sequence in the case of *NUMA1*, fused to the protein of interest. Hela Kyoto cells stably expressing the scFv-GFP were transfected using JetPrime (Polyplus) and a cocktail of three plasmids, including the recombination cassette, and constructs expressing Cas9-HF1 and guide RNAs with an optimized scaffold. Cells were selected on 0.25 μg/ml puromycin for a few weeks. Individual clones were then picked and analyzed by PCR genotyping, fluorescent microscopy, and smiFISH with probes against the SunTag or endogenous mRNA sequence. The sequences targeted by the guide RNAs were (PAM sequences are lowercase): AAGTGAGCCCGACCGAGCGGagg for *ASPM* and GACAGTCACTCCAATGCGCCtgg for *NUMA1*.

**Genotyping**. PCR was done using a Platinum Taq DNA Polymerase (Invitrogen) on genomic DNA prepared with GenElute Mammalian Genomic DNA miniprep (Sigma-Aldrich). The sequences of oligonucleotides are given below. All primers are also listed in Supplementary Table 1.

For genotyping *ASPM*-MS2x24 clones: 5'-TCAGAGGGTATGGAGGGGAA-3' (*ASPM* gene end WT forward) with 5'-GACATCTGTGGCCCTGAAAC-3' (*ASPM* gene end WT reverse) for the WT *ASPM* allele; and 5'-TCAGAGGGTATGGAGGGGAA-3' (*ASPM* gene end forward) with 5'-GCCCTCACATTGCCAAAAGA-3' (IRES reverse) for the edited *ASPM* allele.

For genotyping *NUMA1*-MS2x24 clones: 5'-ACCAAGGACTAAAGGGAGCC-3' (*NUMA1* gene end WT forward) with 5'-CAACCCCACTCCTGAGACAT-3' (*NUMA1* gene end WT reverse) for the WT NUMA1 allele; and 5'-ACCAAGGACTAAAGGGAGCC-3' (*NUMA1* gene end WT forward) with 5'-GCCCTCACATTGCCAAAAGA-3' (IRES reverse) for the edited *NUMA1* allele.

For genotyping *SunTagx32-ASPM* clones: 5'-TGTTCCTGGAAACCGCAATG-3' (*ASPM* gene start WT forward) with 5'-GTTTATGTGTTGTCCCCGCC-3' (*ASPM* gene start WT reverse 1) for the WT *ASPM* gene; and, 5'-AAAAGGGTAGCGGATCAGGA-3' (SunTagx32 forward), with 5'-CATGTGTATGCGTCAAGGGC-3' (*ASPM* gene start reverse) for the edited allele.

For genotyping *SunTagx32-NUMA1* clones: 5'-TCATTGTGCCCCTGGAGATT-3' (*NUMA1* gene start WT forward) with 5'-CAGAGAGACCAGTGCTGTGA-3' (*NUMA1* gene start WT reverse) for the WT *NUMA1* gene; and, 5'-ACCGGTGACTACAAAGACGA-3' (FLAG forward), with 5'-GCTGTGATTCTATGCTGGGC-3' (*NUMA1* gene start reverse) for the edited allele.

**smFISH in low throughput**. Cells grown on glass coverslips or 96-well glass-bottom plates (SensoPlates, Greiner) were fixed for 20 min at RT with 4% paraformaldehyde (Electron Microscopy Sciences) diluted in PBS (Invitrogen), and permeabilized with 70% ethanol overnight at 4 °C.

For smFISH performed on BAC cells, we used a set of 44 amino-modified oligonucleotide probes against the GFP-IRES-Neo sequence present in the BAC construction (sequences given in Supplementary Data 2). Each oligonucleotide probe contained 4 primary amines that were conjugated to Cy3 using the Mono-Reactive Dye Pack (PA23001, GE Healthcare Life Sciences). To this end, the oligos were precipitated with ethanol and resuspended in water. For labeling, 4 μg of each probe was incubated with 6 μl of Cy3 (1/5 of a vial resuspended in 30 μl of DMSO), and 14 μl of carbonate buffer 0.1 M pH 8.8, overnight at RT and in the dark, after extensive vortexing. The next day, 10 μg of yeast tRNAs (Sigma-Aldrich) were added and the probes were precipitated several times with ethanol until the supernatant lost its pink color. For hybridization, fixed cells were washed with PBS and hybridization buffer (15% formamide from Sigma-Aldrich in 1× SSC), and then incubated overnight at 37 °C in the hybridization buffer also containing 130 ng of the probe set for 100 μl of the final volume, 0.34 mg/ml tRNA, 2 mM VRC (Sigma-Aldrich), 0.2 mg/ml RNAse-free bovine serum albumin (BSA) (Roche Diagnostics), and 10% Dextran sulfate (MP Biomedicals). The next day, the samples were washed twice for 30 min in the hybridization buffer at 37 °C and rinsed in PBS. Coverslips were then mounted using Vectashield containing DAPI (Vector Laboratories, Inc.). For smFISH against the MS2 tag, 25 ng of an oligonucleotide labeled by two Cy3 molecules at the first and last thymidine (sequence in Table S2) was used per 100 μl of hybridization mix.

For smiFISH using DNA probes[34], 24–48 unlabeled primary probes were used (sequences given in Supplementary Data 2). In addition to hybridizing to their targets, these probes contained a FLAP sequence that was pre-hybridized to a secondary fluorescent oligonucleotide. To this end, 40 pmoles of primary probes were pre-hybridized to 50 pmoles of the secondary probe in 10 μl of 100 mM NaCl, 50 mM Tris-HCl, 10 mM $MgCl_2$, pH 7.9. Pre-hybridization was performed on a thermocycler with the following program: 85 °C for 3 min, 65 °C for 3 min, and 25 °C for 5 min. The final hybridization mixture contained the probe duplexes (2 μl per 100 μl of final volume), with 1× SSC, 0.34 mg/ml tRNA, 15% Formamide, 2 mM

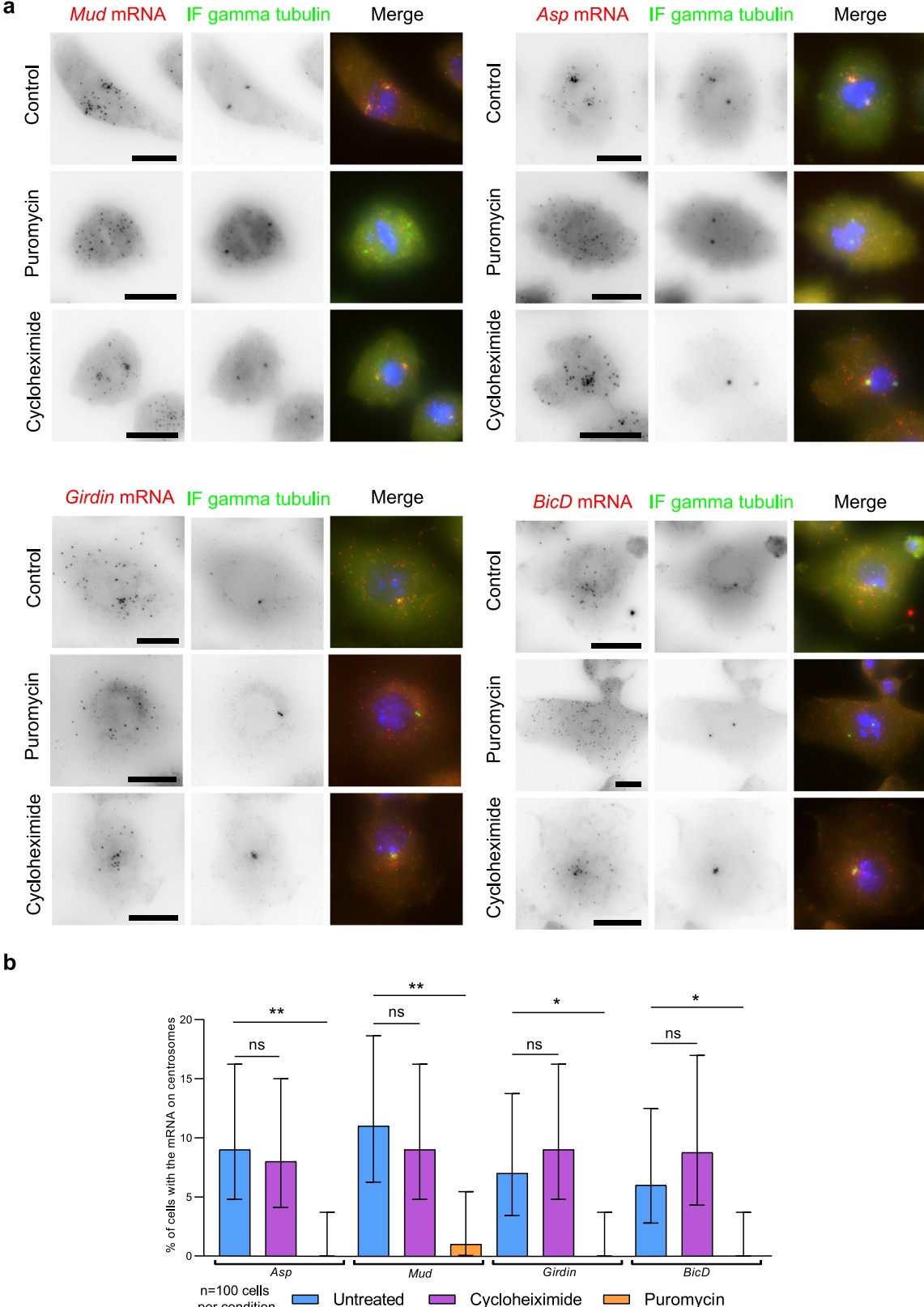

VRC, 0.2 mg/ml RNAse-free BSA, 10% Dextran sulfate. Slides were then processed as above.

**High-throughput smFISH**. To select the gene to be screened, we first included all human genes whose GO Component included one of the following terms: "centrosome", "centriole", "pericentriolar material", "microtubule", "equatorial cell cortex", "midbody", "spindle", "mitotic spindle", "cell division site part",

"kinetochore", "condensed chromosome", "centromere", "telomere". This represented 932 human genes that were manually curated to a final list of 728 genes, which were processed for high-throughput smFISH. Signals were obtained for 602 of the genes. Many genes for which no signal could be obtained are not expressed in HeLa Centrin1–GFP cells.

To perform high-throughput smFISH, a pool of DNA oligonucleotides (GenScript) was used to generate the primary probes. The oligonucleotide design

**Fig. 9 Translation-dependent targeting of centrosomal mRNA is conserved in *Drosophila*. a** Images are micrographs of S2R+ cells treated with either puromycin or cycloheximide, as well as untreated control cells. Left and red: Cy3 fluorescent signals corresponding to centrosomal mRNAs labeled by smiFISH; middle and green: GFP signals corresponding to the gamma-tubulin protein revealed by IF. Blue: DNA stained with DAPI. Scale bar: 10 microns. **b** Bar graph depicting the percentage of interphase cells showing the centrosomal localization of an mRNA after either a puromycin or cycloheximide treatment, as well as control cells. Data were analyzed from 100 cells per condition from two independent experiments and expressed as a percentage of cells with localized mRNA. Binomial proportion 95% confidence intervals are shown in each case and were calculated using the Wilson/Brown method. Statistical significance was evaluated using a two-sided Fisher's exact test. ** indicated a *p* value of <0.01, * a *p* value of <0.05 and ns means nonsignificant.

---

was based on the Oligostan script[34]. The first series of PCR was performed using the oligopool as a template and each of the gene-specific barcoding primers. The second series of PCR was achieved using the following primers: FLAP Y sequence with the addition of the T7 RNA polymerase promoter sequence at its 5′ end (5′ TAATACGACTCACTATAGGGTTACACTCGGACCTCGTCGACATGC ATT-3′), and reverse complement sequence of FLAP X (5′-CACTGAGTCCAGCTCGA AACTTAGGAGG-3′). All PCR reactions were carried out with Phusion DNA Polymerase (Thermo Fisher Scientific), in 96-well plates with a Freedom EVO 200 (Tecan) robotic platform. PCR products were checked by capillary electrophoresis on a Caliper LabChip GX analyzer (PerkinElmer). The products of the second PCR were purified with a NucleoSpin 96 PCR Clean-up kit (Macherey-Nagel), lyophilized, and resuspended in DNase/RNase-free distilled water (Invitrogen). In vitro transcription was subsequently performed with T7 RNA Polymerase and the obtained primary probes were analyzed by capillary electrophoresis using a Fragment Analyzer instrument (Advanced Analytical). Totally, 50 ng of primary probes (total amount of the pool of probes) and 25 ng of each of the secondary probes (TYE 563 labeled LNA oligonucleotides targeting FLAP X and FLAP Y, Qiagen) were pre-hybridized in 100 μL of the following buffer: 1× SSC, 7.5 M urea (Sigma-Aldrich), 0.34 mg/mL tRNA, 10% Dextran sulfate. Pre-hybridization was performed on a thermocycler with the following program: 90 °C for 3 min, 48 °C for 15 min. Glass-bottom 96-well plates with fixed cells in EtOH70% were washed with PBS and hybridization buffer (1× SSC, 7.5 M urea), and the prehybridized mixture og probe was added to each well. Hybridization was then carried out overnight at 48 °C. The next day, plates were washed six times for 10 min in 1× SSC 7.5 M urea at 48 °C. Cells were rinsed with PBS at RT, stained with 1 μg/mL Dapi diluted in PBS, and mounted in 90% glycerol (VWR), 1 mg/mL p-Phenylenediamine (Sigma-Aldrich), PBS pH 8.

Images were acquired on an automated spinning disk microscope (Opera, Perkin Elmer), equipped with a 63× water objective (NA 1.15). Acquisitions were done in 3D with a 600 nm z-step, 16 FOV per well. Two acquisitions were done, one with random fields of view, and one with the field of views centered on mitotic cells, which were selected by a rapid prescan of the wells at low resolution using DAPI staining. Images were processed to generate mosaic multicolor 2D maximum intensity projections that were examined by two experienced microscopists. Localization of mRNA was assessed manually using predefined categories[8], and centrosomal localization was also scored using the Centrin1–GFP label to identify centrosomes.

**Immunofluorescence.** Cells were seeded and fixed as for smFISH. Cells were permeabilized with 0.1% Triton-X100 in PBS for 10 min at room temperature and washed twice with PBS. For centrosome labeling, coverslips were incubated for 1 h at room temperature with a monoclonal anti-γ-tubulin antibody produced in mouse (Sigma-Aldrich, T5326), diluted 1/1000 in PBS. Coverslips were washed twice with PBS, 5 min each time, and incubated with either a FITC (Jackson ImmunoResearch 115-095-062) or Cy5 (Jackson ImmunoResearch 115-176-003) labeled anti-mouse secondary antibody diluted 1/100 in PBS. After 1 h of incubation at RT, coverslips were washed twice with PBS, 5 min each. Coverslips were mounted using Vectashield containing DAPI (Vector Laboratories, Inc.).

**Image analysis.** Mitotic phases were identified based on visual inspection of DNA condensation and cell shape. Early prophase was defined by its low DNA compaction, which increased in late prophase. Early prometaphase was marked by the rupture of the nuclear envelope, while late prometaphase additionally displayed cell rounding. For late mitosis, we subdivided cells into early telophase (without cytokinesis), and late telophase (with cytokinesis marked by the accumulation of HMMR-GFP at cytokinetic bridges). Centrosomal localization was assessed by visual inspection of individual cells.

**Automated analysis of centrosomal mRNA localization.** Nuclei and cell segmentation were performed with Cellpose model[61]. Cells hybridized to reveal endogenous genes were segmented considering a diameter of 140 pixels for the nuclei and 220 pixels for the cells. BAC expressing cells were segmented considering a diameter of 70 pixels for the nuclei and 110 pixels for the cells. Cellpose appears to be highly sensitive to this parameter. Nuclei segmentation was applied to the DAPI channel and cell segmentation to the CellMask or GFP channel. As a post-processing step, we ensure that every nucleus matched with a segmented cell.

Spot detection was performed with Big-FISH[62] (https://github.com/fish-quant/big-fish), a python implementation of FISH-Quant[63]. It starts with a Laplacian of Gaussian filter (LoG) to enhance the spot signal. A local maximal algorithm localizes every peak in the image, then a threshold is applied to discriminate the actual spots from the nonspecific background signal. A dedicated heuristic in Big-FISH automatically sets an optimal threshold. This parameter-free detection pipeline can be scaled to thousands of images. Additional steps included a decomposition of clustered areas where spots cannot be individually identified and detection of foci with a DBSCAN clustering algorithm[64]. mRNA spots were detected from the smFISH channel and centrosomes from the GFP one. Since the GFP centrosome signal is much larger than generic mRNAs spots, we detected them as we would detect mRNA foci but in the GFP channel. Ultimately we retained cells with 1 or 2 detected centrosomes and more than 10 detected mRNAs (54,263 segmented cells in total).

Features were computed with Big-FISH (mRNA spatial features used for the descriptive statistics) and with Cell Cognition (the morphological features used to classify the cell cycle)[65]. Features were computed at the single-cell level based on the detection and segmentation results.

Cell cycle classification was performed with Cell Cognition based on the DAPI channel and the nucleus morphology. Several iterations between automatic classification and manual annotations were done to improve the classifier and refine the annotations.

The code was written in Python. Data manipulation was made with Pandas[66] and NumPy[67]. Visualizations were done with Matplotlib[68] and Seaborn[69]. Computed spatial features and code are accessible at Github https://github.com/Henley13/paper_centrosome_2020 and at https://github.com/fish-quant/big-fish.

**Imaging of fixed cells.** Microscopy slides were imaged on a Zeiss Axioimager Z1 wide-field microscope equipped with a motorized stage, a camera scMOS ZYLA 4.2 MP, using a 63× or 100× objective (Plan Apochromat; 1.4 NA; oil). Images were taken as Z-stacks with one plane every 0.3 μm. The microscope was controlled by MetaMorph and figures were constructed using ImageJ, Adobe Photoshop, Illustrator, and OMERO[70].

Totally, 96-well plates were imaged on an Opera Phenix High-Content Screening System (PerkinElmer), with a 63× water-immersion objective (NA 1.15). Three-dimensional images were acquired, consisting of around 35 slices with a spacing of 0.3 μm.

**Imaging of live cells.** Live imaging was done using a spinning disk confocal microscope (Nikon Ti with a Yokogawa CSU-X1 head) operated by the Andor iQ3 software. Acquisitions were performed using a 100× objective (CF1 PlanApo 1.45 NA oil), and an EMCCD iXon897 camera (Andor).

For fast imaging, we imaged at a rate of at least 1 stack/s for 1–3 min, using stacks with a Z-spacing of 0.4–0.6 μm. This spacing allowed accurate point spread function determination without excessive oversampling. For slow imaging, we collected stacks of around 19 planes with a Z-spacing of 0.6 μm and at a frame rate of one stack every 5 min for 62 h. The power of illuminating light and the exposure time was set to the lowest values that still allowed visualization of the signal. This minimized bleaching, toxicity and maximized the number of frames that were collected. Samples were sequentially excited at 488 and 640 nm in the case of dual-color imaging.

For *ASPM-MS2x24* and *NUMA1-MS2x24*, a time-lapse of a Z-stack covering a 3D section of the cell was acquired in the 488 nm channel while a single Z-stack was acquired in the 640 nm channel to identify the mitotic phase. For mono-color SunTag ASPM and NUMA1 movies, Z-stacks traversing the entire cell were imaged. For dual-color imaging of SunTag-ASPM and MTs, a Z-stack covering a 3D section of the cell was imaged, to maintain high frame rates and compensate for the time required for the second color.

Cells were maintained in an anti-bleaching live-cell visualization medium (DMEM$^{gfp}$; Evrogen), supplemented with 10% FBS at 37 °C in 5% $CO_2$. SiR–DNA (Spirochrome) was kept at 100 nM throughout the experiments to label DNA. SiR–tubulin (Spirochrome) was kept at 100 nM throughout the experiments to label MTs.

**Single-molecule dynamics analysis and single-particle tracking.** The dynamics of *ASPM* mRNAs, and both *NUMA1* mRNAs and polysomes were assessed as follows: mRNAs anchored to the centrosomes as well as those undergoing directed

motion were identified based on visual inspection. The speed of individual mRNA molecules displaying directed motions was measured across at least three frames and calculated using ImageJ.

Single-particle tracking of ASPM polysomes was performed using the TrackMate plugin in ImageJ[71]. The DoG detector was used. Blob diameter was set to 0.7–0.8 microns and the detection threshold was between 100 and 120. Median filtering and sub-pixel localization were additionally used. The simple LAP tracker option was used to construct tracks. Both linking and gap closing distances were assigned a maximum value of 1.5 microns. Three frame gaps were allowed when constructing tracks. Tracks were displayed color-coded according to displacement (red corresponds to highest values while blue corresponds to lowest). For SunTag-ASPM mono-color movies, the top 20 tracks with the highest displacements were chosen. The velocity was calculated by measuring the displacement over at least five frames of directed motion. Directionality toward the centrosome was determined visually. For dual-color movies of SunTag-ASPM and MTs, tracks shorter than 15 s were filtered out. Tracks were classified in one of three categories based on their localization (colocalizing or not to MT, colocalizing with the nuclear envelope), while a fourth category was made for particles showing directed motions.

Tracks were imported and analyzed in R. Instant 1D displacements between frames were calculated along the $x$- and $y$-axis and the resulting histograms were fitted to a Gaussian function, for which variance is directly proportional to the diffusion coefficient ($D$). We also calculated a mean MSD as a function of time, by aligning all tracks at their start and averaging the resulting 2D displacements.

**Reporting summary**. Further information on research design is available in the Nature Research Reporting Summary linked to this article.

## Data availability

Source data are provided with this paper. All relevant data that support the findings of this study are available from the corresponding authors upon reasonable request.

## Code availability

The code used to analyze centrosomal mRNA localization, as well as its raw numerical results are available at https://github.com/Henley13/paper_centrosome_2020 and https://doi.org/10.5281/zenodo.4322750.

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

## Acknowledgements

We thank A. Akhmanova for the *ASPM* CDS, B. Delaval for the HeLa centrin1–GFP cell line, and the MGX and MRI facilities for their technical support. We thank Philippe Fort and Andreas Merdes for their scientific inputs. AS was supported by fellowships from MESRI and FRM. This project was supported by France BioImaging (ANR-10-INBS-04), by the Agence Nationale de la Recherche (ANR-11-BSV8-018-02 and ANR-14-CE10-0018-01), the Fondation pour la Recherche Médicale ("Bioinformatics" grant), the Institut Pasteur, the Ligue Nationale Contre le Cancer, and the Labex EpiGenMed from the framework "Investissements d'avenir". This work was also funded in part by the French government under the management of Agence Nationale de la Recherche as part of the "Investissements d'avenir" program, reference ANR-19-P3IA-0001 (PRAIRIE 3IA Institute).

## Author contributions

The H.T.-smFISH methodology was conceived by E.B. and developed by A.M.T., C.L., E.C., E.B., F.L., M.P., T.G., and V.G. The idea of screening the centrosomal proteome was conceived by E.B., the gene and probe lists were generated by C.L., E.B., and M.P., the screen was conducted by E.C. and F.L. and image data interpretation/curation was performed by E.C., A.S., and E.B. All other investigations were conceived and conducted by A.S., with help from M.C.R. and E.B. for the *GFP-stop-ASPM* CDS experiment, and S.S. for drug treatments and phenotypic counts. A.I., F.M. and T.W. conducted the automated analysis of RNA localization. A.I., A.S., A.M.T., E.C., E.B., F.L., F.M., H.L.H., K.Z., M.C.R., M.P., O.S.K., R.C., T.W., V.G., and X.P. analyzed the data. A.S. prepared the Figures for data visualization. M.P. wrote the HT-smFISH method section, A.S. and E.B. wrote the abstract and discussion, and AS wrote the rest of the paper, which was reviewed and edited by A.S., E.B., H.L.H., K.Z., M.P., O.S.K., and X.P.

## Competing interests

The authors declare no competing interests.
