## [Peer Review File · Nature Communications]

REVIEWER COMMENTS

Reviewer #1 (Remarks to the Author):

Summary and General Comments:

In this manuscript, Safieddine et al. characterize a group of mRNAs that localize to centrosomes and to define on how these mRNAs are transported to these structures. They first present results of a FISH-based screen of 602 mRNAs, from which they identified 8 centrosome-localized mRNAs. Closer observation of these transcripts revealed dynamic and distinctive patterns of localization during interphase and mitosis for these mRNAs and their protein products. The authors then employed comparative translation inhibitor treatments to show that centrosomal targeting of these mRNAs was sensitive to puromycin, suggesting that enrichment to centrosomes requires nascent peptide synthesis. This was further validated for ASPM mRNAs, by inserting a premature stop codon, which also disrupted mRNA localization. Live cell imaging experiments of tagged mRNAs (MS2 system) or nascent protein products (SUNTAG system) revealed distinct populations of ASPM and NUMA1 mRNAs and peptide products with different localization features and kinetics, as well as an intimate association and dependency on microtubules for centrosomal targeting. Lastly, experiments in *Drosophila* orthologs revealed a similar localization pattern and mechanism, which apparently required nascent polypeptides based on puromycin sensitivity.

Overall, this is a high-quality study that is technically elegant and offers significant insights into mRNA targeting to mitotic structures, such as centrosomes. The paper is well written and was easy to follow, with a clearly outlined rationale. Nevertheless, the manuscript would be significantly strengthened by addressing the points outlined below.

Specific comments:

1. The authors should provide more details regarding 602 mRNAs that were screened, how they were selected and the phenotypic counts of the FISH screen presented in Figure 1. This would strengthen the paper.
2. Also regarding Figure 1, the data presented for CCDC88C mRNA, which the authors state is exclusively centrosomally-localized during interphase, is not so clear. There appear to be many CCDC88C mRNA foci and this image is not so convincing in capturing a clear localization to centrosomes.
3. Regarding the data presented with the ASPM variant transgene containing a pre-mature stop codon, did the authors assess whether this transcript variant was expressed to similar levels as the standard GFP-ASPM-CDS mRNA? Stop codon containing transcripts may exhibit altered decay features and it would be good to add RT-qPCR data showing the relative expression levels of both transgenes.
4. The data shows that microtubules are responsible for the transport of mRNAs. The authors state the probable involvement of motor proteins in this process, however they have not shown this directly. Have they assessed whether the localization of the mRNAs is sensitive to the to inhibitors or siRNAs targeting motor protein families? Such data, while not essential, would certainly strengthen the paper.
5. Regarding the data presented in Figures 7E and 7F, what is the percentage of events that appear to involve either sliding on MTs versus MT pulling?

Minor points:

1. The graphs showing quantification should include stars for statistical significance and p-values.

2. Typos:

a. Page 5, line 117. 'All these mRNAs all...'

b. Line 255: C-terminus

c. Line 367: neither

d. At line 455, the authors can conclude on conservation, but not on translation dependency at the point, this statement should be put at the end of the section instead (i.e. line 462).

Reviewer #2 (Remarks to the Author):

The paper by Safieddine et al. investigate the role of RNA localization and local translation in centrosome biology. The authors performed single molecule RNA FISH (smFISH) for human mRNAs (602 genes) coding for centrosomal proteins. They identified 8 transcripts that localize to the centrosomes: ASPM, NUMA1, PCNT, HMMR, BICD2, CCD88C, CEP35 and NIN. From these 8 mRNAs, four are novel candidates that localize to the centrosomes (NIN, CEP350, BICD2, CCDC88). The authors showed that centrosomal genes are trafficking towards the centrosomes in mitosis, and that trafficking requires local active translation and is dependent on microtubules and motors. They also showed that centrosomal mRNA localization varies during the cell cycle, and that prophase is the phase of mitosis where most of these mRNAs localize.

Although the authors address a very novel aspect of RNA localization and local active translation in the context of centrosome biology, similar data on two centrosomal genes (PCNT and ASPM) have already been published in 2018 by Sepulveda et al., *Elife*. Other mRNAs such as HMMR, NUMA1 and ASPM are also part of the bioRxiv preprint by Chouaib et al., 2019. This manuscript would have been stronger if the authors focused on other mRNAs such as NIN, BICD2, CEP350 or CCDC88C rather than ASPM.

In addition, the authors should include some negative control genes that were identified in the screen (Table 1; e.g. centrosomal mRNAs that don't use active translation). The authors should also increase the number of analysed mitotic cells (minimum is n=50 for mitotic cells, instead of n=25) and show single measurements and p-values. These data are missing for every graph in this manuscript.

Major comments

Figure 1: the authors screened for 602 genes in HeLa cells stably expressing Centrin1 (marks centrosomes) and analysed 3D images in interphase and mitotic cells. They found that 8 mRNAs are present at the centrosomes: PCNT, NUMA1, NIN, BICD2, CCD88C and CEP350 (ASPM and NUMA1 are not part of Figure 1). Based on Figure 1B this enrichment is hardly 2-fold between interphase and mitotic cells (prophase). The authors should provide single measurement of mitotic and interphase cells including the p-values. They should also increase the number of mitotic cells (at least 50).

Second paragraph on ASPM, NUMA1 and HMMR is provided in the supplementary methods where the authors used live cell imaging to show that these proteins localize to specific cell cycle stages. I have found very similar results in the Chouaib et al., bioRxiv, 2019, which instead was using fixed cells (Figure 4). The authors cite these two papers (Chouaib et al., 2019, and Poser et al., 2008), but I was not able to find any data on characterisation of HeLa GFP BAC cell lines. Does introduction of ASPM-, HMMR- and NUMA1-BACs affect mitotic timing of HeLa cells?

Figure 2: Using HeLa GFP BAC cell lines for ASPM, NUMA1 and HMMR, the authors showed that mRNA of ASPM, NUMA1 and HMMR localise with their corresponding proteins. Similar result is shown in Figure 4 of Chouaib et al, 2019. The detection of translation factors on centrosomes is an interesting observation but is also shown in a preprint by Chouaib et al, 2019. The authors should either

demonstrate the strong centrosomal localisation of two translation factors or remove “unpublished data” from the manuscript (page 10, line 225).

Figure 3: 6 mRNA were shown to delocalize after puromycin treatment suggesting that their mRNA localisation at the centrosomes requires nascent peptides. In this experiment, the authors should also include untreated cells especially as it was shown in their Figure 3B (quantification). As mentioned above, also Figure 3B should contain single measurements and p-values.

Figure 4: the authors overexpresses ASPM-GFP in HeLa Kyoto cells. As far as I am aware HeLa Kyoto cells already express alpha tubulin-GFP. Are these HeLa Kyoto cells different? What negative control did they use in this experiment (Figure 4A)? Does overexpression of ASPM-GFP affect mitotic timing of HeLa cells?

-To show that the enrichment of these centrosomal mRNAs is dependent on actively translating ribosomes, the authors should perform experiments with emetine and harringtonine.

-What would happen if the authors treat the cells with centrinone? Would polysome still be able to traffic towards the centrosomes?

Figure 6/7: the authors used Suntag system to image nascent peptide of ASPM. They showed that ASPM mRNA is locally translated on microtubules in interphase, and that the polysomes can be anchored to the microtubules.

-In addition to nocodazole treatment, the authors could use Cytochalasin B, which disrupts the actin cytoskeleton. This experiment will demonstrate that microtubules, and not actin filaments, serve as trafficking platform for centrosomal mRNAs to be transported.

Figure 8: They showed that NUMA1 is transported towards centrosomes at the onset of mitosis. Similar observation has been reported for NUMA1 in Chouaib et al., bioRxiv, 2019.

Minor comments

-How did the authors show the specificity of centrosomal FISH probes? Each probe set is usually divided in odd and even probes to show the specificity of their colocalization. Even a single oligonucleotide binding to a highly abundant off-target RNA can lead to spurious signals (please see Cabili et al., 2015, Genome Biology).

- It would be nice if the authors could show that mitosis of HeLa cells expressing centrosomal genes tagged with GFP-BAC, MS2 or Suntag is not affected.

-the authors should also discuss the role of ASPM mRNA localisation at the centrosomes in the context of known role of ASPM in neurogenesis.

Reviewer #3 (Remarks to the Author):

In this study, Safieddine et al. use a high throughput smFISH approach to screen for mRNAs encoding for centrosomal and spindle related proteins in human cells. The authors show that this subset of mRNAs localizes mainly to centrosomes and mitotic spindle microtubules at different stages of cell cycle in a unique fashion mode. By using live single molecule imaging methods the authors also show that these mRNAs may use a local translation mechanism that requires the nascent peptide for their proper localization. They use ASPM and NUMA1 mRNAs as examples to validate these findings. Finally, the authors show that Drosophila orthologous genes also localize onto centrosomes (4 out of 5) in a translation-dependent manner suggesting that local translation by active polysomes at the spindle microtubules is an evolutionary conserved mechanism to ensure proper cell cycle progression.

This is a very elegant work. Congratulations to the authors who put a great amount of effort in demonstrating localization patterns and local translation mechanisms of human centrosomal mRNAs

during cell cycle progression. Although, there is a lack of novelty on the discovery of the presence of RNA in centrosomes and spindle microtubules, overall, the biological insights into the molecular mechanism by this study are significant and potentially interesting to others in the scientific community.

Overall, this is a well-written paper; data/images/movies are clear and well presented. There are, however, a few comments detailed below that could be addressed before recommending for publication.

Major comments

Local RNA translation has been usually thought as the transport of silenced mRNAs "waiting" for the right time and place to be translated. It is worthwhile to point out that in some scenarios an mRNA not only can be de-repressed but also transnationally activated. This is the case of some maternal mRNAs in *Xenopus* oocytes and other developmental model systems where maternal mRNAs are silenced during cell cycle arrest. In this study, Safieddine et al. challenge this view showing that a subset of centrosomal mRNAs in mitotic dividing cells requires active polysome transport to do so. To my knowledge; there is no evidence of this possible mechanism so far. The use of live cell imaging approaches detecting nascent peptide chains allowed this discovery with unprecedented resolution. However, it would be good if the authors could discuss a bit more about the discrepancies among the model systems and also acknowledge the bibliography on RNA localization in meiotic and mitotic *Xenopus* spindles that was not complete and correctly referenced (p 4-5; p24). Blower 2007 and Elisovich 2008 should be properly cited in the text.

Because translation is required for localization, the authors propose that it is the CDS -and not the 3'UTR- that mediates the mechanism of RNA localization. In Figure 4, the authors show that while reporter GFP-ASPM CDS localizes to centrosome, GFP-stop-ASPM CDS does not. However, the authors do not show if the 3'UTR of ASPM mRNA also contribute to the localization. I am still wondering if the 3'UTR of ASPM -or any of the others mRNAs studied here- could also play a role in the dynamic of the spindle assembly through RNA-binding proteins. Could the authors comment if they tested the localization of a reporter RNA containing the 5' and/or 3'UTR of ASPM (without the CDS)? This data would strength the idea that only CDS is sufficient and necessary for proper localization. Also, the authors only show one stage of cell cycle in Figure 4. Does the GFP-ASPM CDS recapitulate the localization of endogenous ASPM during the whole cell cycle? Could the authors perform a colocalization experiment detecting GFP-ASPM CDS and endogenous ASPM? I was also wondering if cell cycle progression is impaired when ASPM mRNAs is mislocalized. Taking advantage of the CRISPR/Cas9 tagging system used in this study, could the authors replace the endogenous ASPM by a mislocalized form of ASPM mRNA (for instance stop-ASPM CDS) and follow the cell cycle progression over time? Is local translation onto centrosomes required to cell cycle progression? What happens with the localization of other centrosomal mRNAs when ASPM is mislocalized? What is the biological relevance of ASPM mRNA localization at centrosomes?

The authors also claim that translation factors such as eIF4E and RPS6 localized at centrosomes in prophase (Chouaib et al (under consideration and unpublished data). How challenging would it be to show a colocalization of these factors with the mRNA and/or the nascent peptides at the centrosomes during cell cycle?

Minor comments

P5, line 117. Please, remove the word 'all' that is repeated in the following sentence: 'All these mRNAs all code for centrosomal proteins...'

P6, line 120. How many genes are 'almost all' human mRNAs? It would be good if the authors can estimate the percentage of human genes encoding for centrosomal proteins so the reader have a better idea of how significative their findings are.

P23, line 516. The authors speculate about a unique program of local translation at the centrosome by these 8 mRNAs encoding for centrosomal proteins. If the nascent peptide is what defines the RNA localization: could the authors discuss if they found, for instance, a common binding motif (similar to the SRP mechanism) among them?

Figure 2 shows RNA localization to centrosome but there is not centrosome labeling as in Figure 1. How did the authors quantify this localization?

Figure 3B. What is the significance of the drug treatments? Please, add p values.

Figure S4, S5 and S6. Please, make clear that puromycin and cycloheximide treatment were present in all conditions by adding a vertical line including the 3 rows of images (mRNA, protein and merge) in panels A and B. Please, do the same thing in Figure S8 panel B.

Figure legend 4 (p36, line 851). Because 'CDS' is used in the text and figure, I would modify the title 'Translation of the ASPM cDNA is necessary and sufficient for its centrosomal localization' for 'Translation of the ASPM CDS is necessary and sufficient for its centrosomal localization'.

Figure 6B. Could the authors comment about that these examples of colocalization are at the nuclear boundary in interphase as well?

Figure 6F. Due to the distribution of the speed data, wouldn't be the median a more accurate measurement than the mean?

Figure legend 7 (p 39, line 925). The text says: '...and the orange one is a track that localizes with MTs.'

However there is no orange arrow depicted in panel B. Did the authors mean blue arrow?

Figure 8E. Due to the distribution of the speed data, wouldn't be the median a more accurate measurement than the mean?

Figure 9B. What is the significance of the drug treatments? Please, add p values.

Figure legend S11F. While the figure legend says: 'White arrowheads indicate the starting position of an mRNA molecules, while blue one follows its position at the indicated time' there are pink and blue arrows in the figure. Please, change accordingly.

Table S1. Could the authors explain what "random" localization means?

Please note that to facilitate reading the revised manuscript, major changes in the main text are highlighted in green.

Reviewer #1 (Remarks to the Author):

Summary and General Comments:

In this manuscript, Safieddine et al. characterize a group of mRNAs that localize to centrosomes and define on how these mRNAs are transported to these structures. They first present results of a FISH-based screen of 602 mRNAs, from which they identified 8 centrosome-localized mRNAs. Closer observation of these transcripts revealed dynamic and distinctive patterns of localization during interphase and mitosis for these mRNAs and their protein products. The authors then employed comparative translation inhibitor treatments to show that centrosomal targeting of these mRNAs was sensitive to puromycin, suggesting that enrichment to centrosomes requires nascent peptide synthesis. This was further validated for ASPM mRNAs, by inserting a premature stop codon, which also disrupted mRNA localization. Live cell imaging experiments of tagged mRNAs (MS2 system) or nascent protein products (SUNTAG system) revealed distinct populations of ASPM and NUMA1 mRNAs and peptide products with different localization features and kinetics, as well as an intimate association and dependency on microtubules for centrosomal targeting. Lastly, experiments in *Drosophila* orthologs revealed a similar localization pattern and mechanism, which apparently required nascent polypeptides based on puromycin sensitivity.

Overall, this is a high-quality study that is technically elegant and offers significant insights into mRNA targeting to mitotic structures, such as centrosomes. The paper is well written and was easy to follow, with a clearly outlined rationale. Nevertheless, the manuscript would be significantly strengthened by addressing the points outlined below.

We thank the reviewer for his/her positive evaluation and we address the points raised below.

Specific comments:

1. The authors should provide more details regarding 602 mRNAs that were screened, how they were selected and the phenotypic counts of the FISH screen presented in Figure 1. This would strengthen the paper.

The 602 genes screened were selected by their GO annotations. We included all human genes whose GO "Cellular component" category included one of the following terms: "centrosome", "centriole", "pericentriolar material", "microtubule", "equatorial cell cortex", "midbody", "spindle", "mitotic spindle", "cell division site part", "kinetochore", "condensed chromosome", "centromere", and "telomere". The last 4 GO terms represented very few genes and they were added to ensure that interesting patterns related to mitosis would not be missed. Altogether, this represents 932

human genes that were manually curated to a final list of 728 genes, and signals were obtained for 602 of them. Many genes for which no signal could be obtained are not expressed in HeLa Centrin1-GFP cells. We thus screened the vast majority of the known human centrosomal proteome. The screen itself was initially screened by manually examining 2D MIP images generated from 3D stacks with an Opera spinning disk microscope (63x oil, 16 fields of view per MIP; this analysis part was done by twice by two experienced microscopists). Only interesting candidates were then confirmed by additional smFISH experiments and manual phenotypic counts. This is now explained in the Material and Methods section.

2. Also regarding Figure 1, the data presented for CCDC88C mRNA, which the authors state is exclusively centrosomally-localized during interphase, is not so clear. There appear to be many CCDC88C mRNA foci and this image is not so convincing in capturing a clear localization to centrosomes.

The localization of CCDC88C mRNA appears less striking since it is expressed less than other centrosomal mRNAs, and the cell we show represents an average scenario where not all mRNA molecules are close to the centrosome.

To better quantify this (and increase the overall number of cells analyzed), we performed automated image analysis on a newly acquired dataset that calculates the proportion of mRNAs within 2 μm of centrosomes. This showed that CCDC88C is indeed enriched at centrosomes, and that this enrichment is sensitive to puromycin, but not cycloheximide (Figure R1 below, please also see Figure 3C-D in the revised manuscript).

Figure R1: Automated quantification of centrosomal mRNA localization. The graph represents the proportion of centrosomal mRNAs in single cells, in the presence and absence of puromycin and cycloheximide treatments. A centrosomal mRNA is defined as a spot localizing within $2\mu\text{m}$ of a centrosome. The box corresponds to the 2nd and 3rd quartiles, the bar is the mean and the red diamond is the median. The whiskers equal 1.5 the Interquartile Range. TRIM59 and TTBK2 are randomly localized mRNAs, while KIF1C and DYNC1H1 are enriched in cellular protrusions and in non-centrosomal foci respectively, and serve as controls. Significance was evaluated with a one sided Welch's *t*-test. ***: nul hypothesis rejected with a 0.1% significance level, and - :null hypothesis not rejected.

3. Regarding the data presented with the ASPM variant transgene containing a pre-mature stop codon, did the authors assess whether this transcript variant was expressed to similar levels as the standard GFP-ASPM-CDS mRNA? Stop codon containing transcripts may exhibit altered decay features and it would be good to add RT-qPCR data showing the relative expression levels of both transgenes.

To test whether differences in expression levels might affect mRNA localization, we did not perform RT-qPCR as suggested, because in transient transfection there is a great cell-to-cell variability in expression levels and we thought that what is important are the expression levels during mitosis and in the very same cells used to score mRNA localization. Therefore, instead of a RT-qPCR, we counted the number of mRNA molecules detected by smFISH in mitotic cells expressing GFP- \pm -stop-ASPM CDS (see Figure R2 below). The results show less than a 0.5 fold reduction that is statistically non-significant when a stop codon is introduced. The differential localization of the two mRNAs is thus unlikely due to differences in expression levels.

*Figure R2: Effect of introducing a stop codon on GFP-ASPM mRNA levels in mitotic cells. (A) A sample image showing how manual detection of single mRNA molecules was performed. Yellow dots indicate single mRNA molecules, while red arrows indicate probe-bound plasmid aggregates resulting from the transient transfection that were excluded from the count. (B) Bar graph showing the mean number of molecules per mitotic cell in HeLa Kyoto cells transiently expressing the indicated plasmid. Error bars represent the standard deviation. Ns is non-significant, two-tailed unpaired *t* test (*p*-value of 0,3274).*

4. The data shows that microtubules are responsible for the transport of mRNAs. The authors state the probable involvement of motor proteins in this process, however they have not shown this directly. Have they assessed whether the localization of the mRNAs is sensitive to inhibitors or siRNAs targeting motor protein families? Such data, while not essential, would certainly strengthen the paper.

To inhibit cytoplasmic Dynein, we performed a Ciliobrevin D treatment for 2 hours at 200 μ M (an inhibitor of the AAA+ ATPase motor cytoplasmic dynein PMID: 22425997). This treatment did not delocalize mRNAs from centrosomes (see Figure R3A below).

To test whether Ciliobrevin D was able to completely inhibit dynein motor activity, we imaged single molecules of mature dynein proteins using a DYNC1H1 SunTag clone (PMID: 27597760).

After a Ciliobrevin D treatment, we could still observe directed movements of dynein-SunTag mature protein molecules, although less frequently. This indicated that Ciliobrevin D does not completely inhibit dynein motor activity (see Figure R3B and R3C below, movies available upon request). The residual activity is likely sufficient to localize mRNAs to centrosomes.

Figure R3: The effect of ciliobrevin D on centrosomal mRNA localization and dynein activity. (A) Images are micrographs of HeLa cells containing a BAC expressing a GFP tagged version of the

gene of interest treated or not with ciliobrevin at 200 μ M for 2 hours. Right and red: Cy3 fluorescent signals corresponding to GFP-tagged mRNAs labeled by smFISH; left and green: GFP signals corresponding to the tagged protein. DNA stained with DAPI. Scale bar: 10 microns. (B) Snapshots of a living SunTagx32-DYNC1H1 cell imaged during interphase with a ciliobrevin treatment. The SunTag signal is shown in black. Scale bar: 10 microns. Time is in seconds. Blue arrowheads indicate the starting position of a single mature protein molecule, while red ones follow its position at the indicated time. (C) Bar graph showing the number of DYNC1H1 motors undergoing directed movements per cell per minute. Data represents the median and error bars indicate a 95% confidence interval. ** indicates a p-value of 0.0021 obtained from a two-tailed unpaired t-test.

5. Regarding the data presented in Figures 7E and 7F, what is the percentage of events that appear to involve either sliding on MTs versus MT pulling?

We initially attempted to quantify this, but then realized that in a large fraction of the movies, classifying pulling and sliding is rather difficult, in large part because of the image quality. Mitotic cells are not only rare but also round and more difficult to image than flat interphasic cells. Moreover, our two color live experiments allow imaging only a fraction of the cell depth (due to imaging speed limits) and thus only a small subset of polysome and microtubule trajectories. Thus while we have clear examples of sliding and pulling, we feel that a quantification is for the moment hazardous and would probably be not reliable.

Minor points:

1. The graphs showing quantification should include stars for statistical significance and p-values.

We added p-values to all existing and new quantifications in the manuscript.

2. Typos:

a. Page 5, line 117. "All these mRNAs all";

b. Line 255: C-terminus

c. Line 367: neither

d. At line 455, the authors can conclude on conservation, but not on translation dependency at the point, this statement should be put at the end of the section instead (i.e. line 462).

We thank the reviewer for pointing these out, and have corrected them.

Reviewer #2 (Remarks to the Author):

The paper by Safieddine et al. investigates the role of RNA localization and local translation in centrosome biology. The authors performed single molecule RNA FISH (smFISH) for human mRNAs (602 genes) coding for centrosomal proteins. They identified 8 transcripts that localize to the centrosomes: ASPM, NUMA1, PCNT, HMMR, BICD2, CCD88C, CEP35 and NIN. From these 8 mRNAs, four are novel candidates that localize to the centrosomes (NIN, CEP350, BICD2, CCDC88). The authors showed that centrosomal genes are trafficking towards the centrosomes in mitosis, and that trafficking requires local active translation and is dependent on microtubules and motors. They also showed that centrosomal mRNA localization varies during the cell cycle, and that prophase is the phase of mitosis where most of these mRNAs localize.

Although the authors address a very novel aspect of RNA localization and local active translation in the context of centrosome biology, similar data on two centrosomal genes (PCNT and ASPM) have already been published in 2018 by Sepulveda et al., *Elife*. Other mRNAs such as HMMR, NUMA1 and ASPM are also part of the bioRxiv preprint by Chouaib et al., 2019. This manuscript would have been stronger if the authors focused on other mRNAs such as NIN, BICD2, CEP350 or CCDC88C rather than ASPM.

We appreciate the reviewer's interest in improving the manuscript. To facilitate reading we have numbered the referee's points below. We would also like to point out a few remarks:

1. The aims of our manuscript are not to document the discovery of centrosomal mRNAs, but to: (i) perform a high-throughput smFISH screen dedicated to the **human centrosomal proteome** to find all human centrosomal mRNAs, and (ii) detail the **mechanism of localization** and assess its generality.

2. Although they provide evidence of translation-dependent localization, neither Sepulveda et al., nor Chouaib et al., (now published as PMID: 32783880) provide definitive proof of polysome targeting to centrosomes (or to any other subcellular location). In fact, Chouaib et al concludes by raising the two questions: (i) "*In the future, it will be interesting to determine....and whether the polysomal complex reaches destination via diffusion or active transport*" and (ii) "*cells may have dedicated systems to transport polysomes*". This manuscript is the first to demonstrate **active targeting of polysomes by motors**, directly answering such questions.

3. This work details the cell cycle dependent localization of 4 known and 4 new centrosomal RNAs, including across mitotic subphases. This was not done in Chouaib et al., and it demonstrates the existence of a dynamic translational program operating on centrosomes (i.e. specific RNAs translating at centrosomes at precise times). This concept was purely speculative in the discussion section of Chouaib et al.: "*it is tempting to speculate that there is a specific translational program that takes place at centrosomes*".

4. This manuscript directly demonstrates that human centrosomal mRNAs are conserved in *Drosophila* in terms of mRNA identity, localization and transport mechanism. To our knowledge,

this is the first study reporting conservation of mRNA localization and mechanisms in such distant species.

5. We chose ASPM as a candidate for MS2 and SunTag tagging due to the fact that it **uniquely localizes to centrosomes across all mitotic phases**. Indeed, this facilitated imaging mitotic mRNA and polysome dynamics due to the larger imaging time window. Our second candidate was NUMA1 simply due to the extensive literature available on this gene, which will broaden our audience.

1-In addition, the authors should include some negative control genes that were identified in the screen (Table 1; e.g. centrosomal mRNAs that don't use active translation). The authors should also increase the number of analysed mitotic cells (minimum is n=50 for mitotic cells, instead of n=25) and show single measurements and p-values. These data are missing for every graph in this manuscript.

We agree that adding more controls would strengthen our manuscript. However, our screen did not reveal any centrosomal mRNAs that do not use active translation for localizing. Thus, we added in our quantifications two mRNAs (TRIM59, and TTBK2) that are randomly localized, as well as KIF1C that is enriched in cellular protrusions (away from centrosomes), and DYNC1H1 that localizes in perinuclear foci (please see Figure R1 below and Figure 3 in the revised manuscript).

Concerning quantifications and number of cells, we would like to point out two things:

1. We acquired an extensive new dataset of images for all 8 mRNAs (untreated, or treated with either puromycin or cycloheximide) covering more than 50,000 cells. We developed an automated image analysis pipeline that measures centrosomal mRNA enrichment using a set of features. In brief, this consisted of segmenting cells, detecting single mRNA molecules, and calculating the proportion of RNAs detected within 2 μm of a centrosome. This drastically increased the number of cells analyzed and provided single cell measurements as the reviewer requested. This approach nicely complements the manual phenotypic counts that were already in the paper.
2. The counts already included in the paper correspond to two independent experiments, each counting 25 cells per mitotic phase. We now combined this with the new data set and obtained between 50-70 cells in each mitotic phase, and in each condition (Figures 2, S4-S6). We performed Fisher's exact test for all appropriate data, and modified how the graphs and legends are presented to reflect this.

Major comments

2-Figure 1: the authors screened for 602 genes in HeLa cells stably expressing Centrin1 (marks centrosomes) and analysed 3D images in interphase and mitotic cells. They found that 8 mRNAs are present at the centrosomes: PCNT, NUMA1, NIN, BICD2, CCD88C and CEP350 (ASPM and NUMA1 are not part of Figure 1). Based on Figure 1B this enrichment is hardly 2-fold between interphase and mitotic cells (prophase). The authors should provide single measurement of mitotic and interphase cells including the p-values. They should also increase the number of mitotic cells (at least 50).

- By stating in the main text states that “the bulk of centrosomal mRNAs localize to centrosomes most strongly during prophase”, we mean that most centrosomal mRNAs species that we found localize during prophase. We made this sentence clearer by removing “most strongly”.
- The automated image analysis of the new dataset significantly increases the overall number of cells analyzed, as well as adds a more quantitative aspect (including single cell measurements of mRNA localizations features; see previous point). As the referee points out, not all the mRNAs molecules localize to centrosomes. This appears to be common in cell lines (see Chouaib et al., Dev Cell, 2020). It should also be noted that the translational status of the non-localizing mRNAs is not known.
- We changed how we present the data to better reflect 50-70 cells we counted in each condition and mitotic phase, and added p-values.

3-Second paragraph on ASPM, NUMA1 and HMMR is provided in the supplementary methods where the authors used live cell imaging to show that these proteins localize to specific cell cycle stages. I have found very similar results in the Chouaib et al., bioRxiv, 2019, which instead was using fixed cells (Figure 4).

In Chouaib et al., protein localization was analyzed only in fixed cells. The objective of Supplementary Figure 1 is to follow the localization of the three proteins across a full cell cycle and for this we used live cell imaging. This revealed for instance that ASPM protein levels rise gradually throughout the cell cycle, culminating before mitosis, and that HMMR-GFP localizes to centrosomes only a few hours before mitotic entry. Thus, when we see HMMR mRNA co-localizing with its protein on centrosomes (Figure 2), we know when in the cell cycle this happens. Such observations could not be made on fixed cells in Chouaib et al. (PMID: 32783880).

4-The authors cite these two papers (Chouaib et al., 2019, and Poser et al., 2008), but I was not able to find any data on characterisation of HeLa GFP BAC cell lines. Does introduction of ASPM-, HMMR- and NUMA1-BACs affect mitotic timing of HeLa cells?

To test the effect of BAC introduction on mitotic progression (and the cell cycle in general), we performed a cell cycle profile analysis using flow cytometry. We compared the fraction of cells in S/G2M/G0G1 to that of HeLa Kyoto (the parental cell line in which the BACs were inserted). Results are presented in Figure R4 below and show comparable BAC cell cycle profiles with respect to the parental HeLa Kyoto line.

Figure R4: Cell cycle profiles of the BAC cell lines used in this study obtained using flow cytometry. Graphs are histograms representing the intensity of a DNA marker (propidium iodide). Black dotted line represents the whole population of single cells (gated to remove debris and doublets). Red area represents cells in G0G1, while yellow represents S, and blue is G2M. The percentage of cells in each phase is shown in the adjacent tables.

5-Figure 2: Using HeLa GFP BAC cell lines for ASPM, NUMA1 and HHMR, the authors showed that mRNA of ASPM, NUMA1 and HHMR localise with their corresponding proteins. Similar result is shown in Figure 4 of Chouaib et al, 2019. The detection of translation factors on centrosomes is an interesting observation but is also shown in a preprint by Chouaib et al, 2019. The authors should either demonstrate the strong centrosomal localisation of two translation factors or remove "unpublished data" from the manuscript (page 10, line 225).

Although these 3 mRNAs were described as centrosomal in Chouaib et al., (as mentioned in the text of the present manuscript) the cell cycle dependent localization was not examined in detail, and in particular not during the different mitotic stages. Here, we found that ASPM uniquely localizes during all 5 mitotic phases, while NUMA1 localizes only during early mitosis (prophase

and prometaphase), and HMMR during early and late mitosis (prophase, prometaphase, and telophase). Combining this with similar observations for 5 other mRNAs provided solid evidence for a centrosomal translation program. Moreover, this study shows that these three transcripts localize to distinct peri-centrosomal regions (Figures S2 and S3).

We attempted to co-detect translation factors with centrosomal mRNAs. However, these particular antibodies were not compatible with our smFISH hybridization conditions. Since Chouaib et al., already contains immunofluorescence data on translation factors, we decided to remove this sentence from the paper (page 10, line 225 of the original paper).

6-Figure 3: 6 mRNA were shown to delocalize after puromycin treatment suggesting that their mRNA localisation at the centrosomes requires nascent peptides. In this experiment, the authors should also include untreated cells especially as it was shown in their Figure 3B (quantification). As mentioned above, also Figure 3B should contain single measurements and p-values.

We included untreated cells in revised Figure 3A and p-values were added to Figure 3B (and all other appropriate graphs in the paper). Moreover, we expanded this figure by adding quantifications resulting from our automated image analysis pipeline applied to a new dataset (which includes > 50,000 cells; please see details above in point 1 of this Referee, revised Figure 3C & 3D, and the main text).

7-Figure 4: the authors overexpress ASPM-GFP in HeLa Kyoto cells. As far as I am aware HeLa Kyoto cells already express alpha tubulin-GFP. Are these HeLa Kyoto cells different? What negative control did they use in this experiment (Figure 4A)? Does overexpression of ASPM-GFP affect mitotic timing of HeLa cells?

We used HeLa Kyoto cells originating from the Hyman lab (Poser et al., Nat Methods, 2008). These cells do not express any GFP marker.

Concerning the negative control, we used a HeLa cell line stably expressing Centrin 1-GFP (a centrosomal protein encoded from a non-centrosomal mRNA) and performed smFISH against the GFP tag. As expected, we did not see an enrichment of the Centrin 1-GFP on centrosomes (Figure R5 below and Figure 4 in the revised manuscript for the full set of experiments).

To assess the effect of ASPM-GFP on the cell cycle, we did not use transient transfection because of the low number of transfected cells and the cell cycle perturbation possibly induced by the transfection reagents, but performed a FACS analysis with the ASPM-BAC cells (which is also overexpressed). The data indicate that the cells progress normally through the cell cycle, as mentioned above in the point 4 of the same referee (see Figure R4 above).

Figure R5: Intracellular distribution of Centrin 1-GFP mRNAs. Images are micrographs of HeLa cells stably expressing Centrin1-GFP. Left and red: single molecules of Centrin 1-GFP revealed by smFISH against the GFP sequence. Middle and green: GFP signal corresponding to centrosomes labeled by Centrin 1-GFP. Scale bar is 10 microns.

8-To show that the enrichment of these centrosomal mRNAs is dependent on actively translating ribosomes, the authors should perform experiments with emetine and harringtonine.

We performed the suggested experiments. Treating cells with 30 minutes with harringtonine disrupted mRNA targeting to centrosomes, while a 30 minute treatment with emetine had no effect (see Figure R6 and R7 below). This was observed with all 8 mRNAs and reinforces with our model that the nascent peptide is required for localizing the mRNA.

Figure R6: The effects of Harringtonin and Emetine on the localization of BAC-transcribed ASPM, NUMA1, and HMMR mRNAs. (A) Micrographs of HeLa cells expressing ASPM-GFP, NUMA1-GFP or HMMR-GFP BACs, treated with either harringtonine or emetine for 30 minutes, and imaged during mitosis (all phases for ASPM, while only early mitosis for NUMA1 and HMMR). Middle and red: Cy3 fluorescent signals corresponding to BAC mRNAs labeled by smFISH with probes against the GFP RNA sequence; left and green: GFP signals corresponding to the protein of interest. Blue: DNA stained with DAPI. Scale bar: 10 microns. (B) Bar graph depicting the percentage of mitotic cells showing a centrosomal localization of the three mRNAs before or after the indicated treatment (n=50 cells per condition, counted from two independent

experiments). Error bars represent a 95% confidence interval. Statistical significance was evaluated using a two-sided Fisher's exact test. **** indicates a p-value of <0.0001, ns is non-significant

Figure R7: The effects of Harringtonin and Emetine on the localization of endogenous mRNAs. (A) Micrographs of HeLa cells expressing Centrin 1-GFP imaged during interphase. Middle and

*red: Cy3 fluorescent signals corresponding to endogenous mRNAs labeled by smFISH; left and green: GFP signals corresponding to the Centrin 1-GFP signal . Blue: DNA stained with DAPI. Scale bar: 10 microns. (B) Bar graph depicting the percentage of cells showing a centrosomal localization of the five mRNAs before or after the indicated treatment (n=50 cells per condition, counted from two independent experiments). Treatments were for 30 minutes. Error bars represent a 95% confidence interval. Statistical significance was evaluated using a two-sided Fisher's exact test. **** indicates a p-value of <0.0001, ns is non-significant*

9-What would happen if the authors treat the cells with centrinone? Would polysomes still be able to traffic towards the centrosomes?

Centrinone is a Plk4 inhibitor that prevents centrosome maturation. We treated cells with centrinone for various periods and up to 3 days of continuous treatment at 125nM (further treatment led to apoptosis and a sharp decline in mitotic cells). Despite this, we were still able to detect centrioles labeled with an anti γ -tubulin immunofluorescence. SmFISH on treated cells shows that centrosomal mRNAs do indeed localize after a 3 day centrinone treatment (see Figure R8 below).

Figure R8: The effects of centrinone on the localization of ASPM and NUMA1 mRNAs. (A) Micrographs of HeLa Kyoto cells treated with centrinone for 3 consecutive days at 125nM. Middle and red: Cy3 fluorescent signals corresponding to ASPM or NUMA1 mRNAs labeled with smFISH

against the endogenous mRNAs; left and green: Cy5 signals corresponding to an anti γ -tubulin immunofluorescence. Blue: DNA stained with DAPI. Scale bar: 10 microns. (B) Bar graph depicting the percentage of mitotic cells showing a centrosomal localization of the two mRNAs before or after the centrinone treatment (n=50 cells per condition, counted from two independent experiments). Error bars represent a 95% confidence interval. Statistical significance was evaluated using a two-sided Fisher's exact test. Ns is non-significant.

10-Figure 6/7: the authors used Suntag system to image nascent peptide of ASPM. They showed that ASPM mRNA is locally translated on microtubules in interphase, and that the polysomes can be anchored to the microtubules. In addition to nocodazole treatment, the authors could use Cytochalasin B, which disrupts the actin cytoskeleton. This experiment will demonstrate that microtubules, and not actin filaments, serve as trafficking platform for centrosomal mRNAs to be transported.

We treated cells with cytochalasin D at 1 μ g/ml for 2 hours and co-labeled actin filaments to ensure that the drug efficiently depolymerizes actin (see Figure R9A below). We then performed smFISH experiments which revealed centrosomal enrichment of all the 8 centrosomal mRNAs after a cytochalasin treatment (see Figure R9 below). This indicates that microtubules, but not actin filaments are required for centrosomal mRNA trafficking.

Figure R9: The effects of cytochalasin D on the localization of centrosomal mRNAs. (A) Micrographs of HeLa Kyoto cells treated with cytochalasin D for 2 hours at 1 μ g/ml. Middle and red: Cy5 signal corresponding to a Phalloidin-TexasRed stain. Blue represents a DAPI stain. Scale bar is 10 microns. **(B)** Micrographs of HeLa Kyoto cells treated or not with cytochalasin D (same dose and duration as in panel A). Middle and red: Cy3 fluorescent signals corresponding

to ASPM or NUMA1 mRNAs labeled with smFISH against the endogenous mRNAs ; left and green: Cy5 signals corresponding to an anti γ -tubulin immunofluorescence. Blue: DNA stained with DAPI. Scale bar: 10 microns. (C) Bar graph depicting the percentage of cells showing a centrosomal localization of all 8 mRNAs before or after the cytochalasin D treatment (n=50 cells per condition, counted from two independent experiments). Early mitotic cells were counted for ASPM, NUMA1, and HMMR; while interphasic cells were counted for all other transcripts. Error bars represent a 95% confidence interval. Statistical significance was evaluated using a two-sided Fisher's exact test. Ns is non-significant.

11-Figure 8: They showed that NUMA1 is transported towards centrosomes at the onset of mitosis. Similar observation has been reported for NUMA1 in Chouaib et al., bioRxiv, 2019.

Chouaib et al. simply states that NUMA1 mRNAs are not localized in interphase, but accumulate on centrosomes during mitosis (without specifying which phases, or the differential localization compared to other centrosomal mRNAs). More importantly, the mechanism of localization was not addressed. In Figure 7 of this manuscript we now :

1. Combined the MS2 and SunTag systems with gene editing to image endogenous NUMA1 mRNAs and polysomes in living mitotic cells.
2. Documented active transport of NUMA1 mRNAs toward centrosomes and more importantly, NUMA1 polysomes.
3. Showed that active polysome targeting mechanisms exist not just for ASPM, but also for other centrosomal transcripts, suggesting widespread usage of this trafficking mechanism.

Minor comments

12-How did the authors show the specificity of centrosomal FISH probes? Each probe set is usually divided in odd and even probes to show the specificity of their colocalization. Even a single oligonucleotide binding to a highly abundant off-target RNA can lead to spurious signals (please see Cabili et al., 2015, Genome Biology)

It is true that a single probe binding an abundant off-target RNA can produce unwanted signals. However this likely manifests as background fluorescence rather than single molecules. The case of Cabili et al. is particular since they hybridized lncRNAs that contain repeats. Such repeats can lead to the appearance of single molecules even with single probes.

Nevertheless, our manuscript contain several instances that demonstrate the specificity of our probesets:

- Endogenous ASPM and NUMA1 probes co-localize with MS2 probes in Figures 4B and 8A where the genes have been edited to contain MS2 repeats (note that not all molecules co-localize since the ASPM-MS2 and NUMA1-MS2 clones are heterozygous).
- GFP IRES Neo probes labeling of BAC tagged transcripts reveal centrosome localization with the same temporal patterns as probes targeting the endogenous genes (example: ASPM BAC mRNA in Figure 2 A and endogenous ASPM mRNA in Figure 5B and 6B).
- Several mRNAs were analyzed with several probe sets with identical results (ASPM, NUMA1, HMMR, NIN, BICD2). Also, note that several mRNAs localizes in both human and Drosophila cells (NIN, BICD2, CCDC88C, NUMA1, ASPM).

13- It would be nice if the authors could show that mitosis of Hela cells expressing centrosomal genes tagged with GFP-BAC, MS2 or Suntag is not affected.

We performed cell cycle analysis using flow cytometry for all cell lines used in this study. The BAC cell lines can be found above (Fig. R4) and the ASPM/NUMA1 MS2 and SunTag below (Fig R10):

Figure R10: Cell cycle profiles of the MS2 and SunTag cell lines used in this study obtained using flow cytometry. Graphs are histograms representing the intensity of a DNA marker (propidium iodide). Black dotted line represents the whole population of single cells (gated to remove debris and doublets). Red area represents cells in G0G1, while yellow represents S, and blue is G2M. The percentage of cells in each phase is shown in the adjacent tables.

14-the authors should also discuss the role of ASPM mRNA localisation at the centrosomes in the context of known role of ASPM in neurogenesis.

We added a paragraph explaining this in the discussion.

Reviewer #3 (Remarks to the Author):

In this study, Safieddine et al. use a high throughput smFISH approach to screen for mRNAs encoding for centrosomal and spindle related proteins in human cells. The authors show that this subset of mRNAs localizes mainly to centrosomes and mitotic spindle microtubules at different stages of cell cycle in a unique fashion mode. By using live single molecule imaging methods the authors also show that these mRNAs may use a local translation mechanism that requires the nascent peptide for their proper localization. They use ASPM and NUMA1 mRNAs as examples to validate these findings. Finally, the authors show that *Drosophila* orthologous genes also localize onto centrosomes (4 out of 5) in a translation-dependent manner suggesting that local translation by active polysomes at the spindle microtubules is an evolutionary conserved mechanism to ensure proper cell cycle progression.

This is a very elegant work. Congratulations to the authors who put a great amount of effort in demonstrating localization patterns and local translation mechanisms of human centrosomal mRNAs during cell cycle progression. Although, there is a lack of novelty on the discovery of the presence of RNA in centrosomes and spindle microtubules, overall, the biological insights into the molecular mechanism by this study are significant and potentially interesting to others in the scientific community.

We are very glad the reviewer appreciated the manuscript.

Overall, this is a well-written paper; data/images/movies are clear and well presented. There are, however, a few comments detailed below that could be addressed before recommending for publication.

Major comments

Local RNA translation has been usually thought as the transport of silenced mRNAs, waiting for the right time and place to be translated. It is worthwhile to point out that in some scenarios an mRNA not only can be de-repressed but also transnationally activated. This is the case of some maternal mRNAs in *Xenopus* oocytes and other developmental model systems where maternal mRNAs are silenced during cell cycle arrest. In this study, Safieddine et al. challenge this view showing that a subset of centrosomal mRNAs in mitotic dividing cells requires active polysome transport to do so. To my knowledge; there is no evidence of this possible mechanism so far. The use of live cell imaging approaches detecting nascent peptide chains allowed this discovery with unprecedented resolution. However, it would be good if the authors could discuss a bit more about the discrepancies among the model systems and also acknowledge the bibliography on RNA localization in meiotic and mitotic *Xenopus* spindles that was not complete and correctly referenced (p 4-5; p24). Blower 2007 and Eliscovich 2008 should be properly cited in the text.

We thank the review for pointing out the mechanistic insight gained from our live imaging approaches.

We added a paragraph in the introduction highlighting the bibliography the reviewer mentioned.

Because translation is required for localization, the authors propose that it is the CDS -and not the 3'UTR- that mediates the mechanism of RNA localization. In Figure 4, the authors show that while reporter GFP-ASPM CDS localizes to centrosome, GFP-stop-ASPM CDS does not. However, the authors do not show if the 3'UTR of ASPM mRNA also contributes to the localization. I am still wondering if the 3'UTR of ASPM -or any of the others mRNAs studied here- could also play a role in the dynamic of the spindle assembly through RNA-binding proteins. Could the authors comment if they tested the localization of a reporter RNA containing the 5' and/or 3'UTR of ASPM (without the CDS)? This data would strengthen the idea that only CDS is sufficient and necessary for proper localization.

We cloned a firefly luciferase reporter containing the ASPM 3'UTR and performed smFISH against the firefly sequence to detect mRNAs transcribed from this reporter. SmFISH revealed that mRNAs transcribed from this reporter carrying the ASPM 3'UTR do not localize to centrosomes (Figure R11 below and Figure 4 of the revised manuscript). This indicates that the ASPM 3'UTR by itself cannot target its mRNA to centrosomes, and it provides a nice addition to main Figure 4.

Figure R11: Intracellular distribution of Firefly luciferase ASPM 3'UTR mRNAs. Images are micrographs of HeLa cells transiently expressing firefly luciferase fused to the ASPM 3'UTR. Left and red: single molecules of firefly luciferase mRNA revealed by smFISH against the firefly luciferase sequence. Middle and green: Cy5 signal corresponding to centrosomes labeled by anti γ -tubulin immunofluorescence. The DAPI DNA stain is shown in blue. Scale bar is 10 microns.

Could the authors perform a colocalization experiment detecting GFP-ASPM CDS and endogenous ASPM?

We performed this experiment and observed a co-localization of signals coming from the probes recognizing endogenous ASPM, and those targeting the GFP sequence (GFP-ASPM CDS mRNA, Figure R12 below). However, we would like to point out that the probes targeting the endogenous ASPM also inevitably target the exogenous GFP-ASPM transcript due to the common ASPM sequences. Thus, the endogenous mRNA signal will always also correspond to the exogenous one as well.

Figure R12: Co-detecting endogenous and exogenous ASPM mRNA. Images are micrographs of HeLa cells transiently expressing GFP-ASPM CDS. Left and red: single molecules of GFP-ASPM CDS mRNA revealed by smFISH against the GFP sequence. Middle and green: GFP signal corresponding to the mature ASPM protein. Right and blue: single molecules of endogenous and exogenous ASPM mRNA revealed by smFISH against the ASPM sequence itself.

I was also wondering if cell cycle progression is impaired when ASPM mRNAs is mislocalized. Taking advantage of the CRISPR/Cas9 tagging system used in this study, could the authors replace the endogenous ASPM by a mislocalized form of ASPM mRNA (for instance stop-ASPM CDS) and follow the cell cycle progression over time? Is local translation onto centrosomes required to cell cycle progression? What happens with the localization of other centrosomal mRNAs when ASPM is mislocalized? What is the biological relevance of ASPM mRNA localization at centrosomes?

We thank the reviewer for these suggestions. We believe that a Stop-ASPM would not answer the question because it would be analogous to a knock-out without any protein expressed. We did try to use the ASPM-MS2x24 and an MCP with a plasma membrane localization sequence to tether the ASPM mRNA to the membrane and prevent its centrosomal localization. This approach however did not work and the mRNA was still centrosomal in such cells. Therefore, while this series of experiments is indeed interesting, we believe that finding a successful strategy to delocalize ASPM mRNA is a full project in itself and goes beyond the timescale and scope envisioned for a revision. We also point out that the revised manuscript already includes a large amount of data (9 main Figures and 12 Supplementary Figures, 3 Tables).

The authors also claim that translation factors such as eIF4E and RPS6 localized at centrosomes in prophase (Chouaib et al (under consideration and unpublished data). How challenging would it be to show a colocalization of these factors with the mRNA and/or the nascent peptides at the centrosomes during cell cycle?

We attempted this several times, but the antibodies recognizing translation factors are not compatible with the smFISH hybridization conditions. Since Chouaib et al., already contains data describing translation factor localization, we simply removed this notion from the text.

Minor comments

P5, line 117. Please, remove the word "all" that is repeated in the following sentence: "All these mRNAs all code for centrosomal proteins"

We thank the reviewer for pointing this out and removed the duplicated word.

P6, line 120. How many genes are "almost all" human mRNAs? It would be good if the authors can estimate the percentage of human genes encoding for centrosomal proteins so the reader have a better idea of how significant their findings are.

We screened 728 genes coding for putative centrosomal proteins and obtained signals for 602 of them. We included all human genes whose GO term in the "Component" category included one of the following term: "centrosome", "centriole", "pericentriolar material", "microtubule", "equatorial cell cortex", "midbody", "spindle", "mitotic spindle", "cell division site part", "kinetochore", "condensed chromosome", "centromere", "telomere". This represents 932 human genes and the list was manually curated to a final list of 728 genes. Based on these numbers, we estimate that we screened 65-82 % of the centrosomal proteome. Note that for most of the 126 genes for which we obtained no signals, the mRNA is simply not expressed in HeLa cells.

P23, line 516. The authors speculate about a unique program of local translation at the centrosome by these 8 mRNAs encoding for centrosomal proteins. If the nascent peptide is what defines the RNA localization: could the authors discuss if they found, for instance, a common binding motif (similar to the SRP mechanism) among them?

This is a very interesting point. We performed a protein alignment of both human and Drosophila proteins encoded from centrosomal mRNAs. We found several SMC (structural maintenance of chromosomes) domains in a number of proteins (HMMR, CEP350, NUMA1, PCNT, NIN).

Interestingly, these domains were also found in some of the *Drosophila* orthologs: Plp, Mud and Bsg25D. This suggests the involvement of this domain in centrosomal targeting for a subset of these mRNAs (see Figure R12 below), but it would require additional experiments to prove it.

Figure R12: Protein alignments of sequences encoded from centrosomal localized mRNAs. Names in red indicate proteins that accumulate on centrosomes.

Figure 2 shows RNA localization to the centrosome but there is not centrosome labeling as in Figure 1. How did the authors quantify this localization?

We used the mature GFP-tagged ASPM, NUMA1, and HMMR centrosomal proteins as a label. This was clarified in the revised text.

In the newly acquired dataset, an additional immunofluorescence against γ -tubulin was performed. In these cases, it was necessary to properly pinpoint the location of the centrosome in order to quantify the amount of mRNAs present within a 2 μ m diameter circle centered on the centrosome.

Figure 3B. What is the significance of the drug treatments? Please, add p values.

We added p-values on all graphs in the manuscript.

Figure S4, S5 and S6. Please, make clear that puromycin and cycloheximide treatment were present in all conditions by adding a vertical line including the 3 rows of images (mRNA, protein and merge) in panels A and B. Please, do the same thing in Figure S8 panel B.

We improved the clarity of the mentioned figures.

Figure legend 4 (p36, line 851). Because "CDS" is used in the text and figure, I would modify the title "Translation of the ASPM cDNA is necessary and sufficient for its centrosomal localization"; for "Translation of the ASPM CDS is necessary and sufficient for its centrosomal localization"

We apologize for this textual error and have corrected it.

Figure 6B. Could the authors comment about that these examples of colocalization are at the nuclear boundary in interphase as well?

We assume the reviewer is referring to co-localization of ASPM mRNA and polysomes at the nuclear envelope. It has been reported previously that ASPM mRNAs are translated at the nuclear pore and remain attached there for about 20 mins during interphase (Chouaib et al. 2020). Such local translation could help in targeting the ASPM mature protein to the nucleoplasm.

Figure 6F. Due to the distribution of the speed data, wouldn't be the median a more accurate measurement than the mean?

We included the median in the graph.

Figure legend 7 (p 39, line 925). The text says: "and the orange one is a track that localizes with MTs". However there is no orange arrow depicted in panel B. Did the authors mean blue arrow?

We apologize for this textual error and have corrected it.

Figure 8E. Due to the distribution of the speed data, wouldn't be the median a more accurate measurement than the mean?

We included the median in the graph.

Figure 9B. What is the significance of the drug treatments? Please, add p values.

We added p-values.

Figure legend S11F. While the figure legend says: "White arrowheads indicate the starting position of an mRNA molecules, while blue one follows its position at the indicated time"; there are pink and blue arrows in the figure. Please, change accordingly.

We apologize for this textual error and have corrected it.

Table S1. Could the authors explain what "random" localization means?

By random we mean an mRNA that is not particularly enriched at any subcellular location. This term becomes clearer after we include two "randomly" localized mRNAs in our automated analysis (please see revised Figure 3). Perinuclear mRNAs in this table are enriched around the nucleus. However, since they encode secreted or membrane proteins, they likely correspond to translation on the ER.

REVIEWERS' COMMENTS

Reviewer #1 (Remarks to the Author):

I am satisfied with the significant efforts put forth by the authors to address reviewer concerns and now recommend acceptance of the manuscript.

Reviewer #2 (Remarks to the Author):

The authors added a significant amount of new experiments, which strongly improved the manuscript, in particular new Figure 3. I have few comments regarding the new figures that would be good to discuss and address:

- 1) Fig R4: there is a slight increase in G2/M population of cells in NUMA1 BAC cells, which is also obvious from the cell cycle profile. This suggests that introduction of BAC for NUMA1 might affect mitosis (24% Bac NUMA1 vs 19% controls).
- 2) Fig R6A: It seems like the metaphase cells showed in this image after emetine is bigger compared to the control metaphase. Are you sure the scale is the same? There is also a typo error, it should be written emetine, not ementine.
- 3) Fig R8: the authors treated the cells with centrinone and concluded that there is no effect of the centrinone on centrosomal mRNA localisation. Again, representing the cells in the same stage of mitosis would be better despite the fact that ASPM mRNA levels are similar from prophase to anaphase (Fig 2 A, B). The same for NUMA1 mRNA, as its mRNA is high in prophase but goes down in prometaphase (Fig 2 C, D). From the representative images it looks like that centrinone reduced the signal of NUMA1 and ASPM mRNA on the centrosome.
- 4) Fig R9B: It would be better to show the same stage (metaphase) for ASPM mRNA after Cytochalasin D treatment. The authors should clarify what are the early mitotic cells counted for this graph as they stated on page 10 for Figure 2 : "Here, we found that ASPM uniquely localizes during all 5 mitotic phases, while NUMA1 localizes only during early mitosis (prophase and prometaphase), and HMHR during early and late mitosis (prophase, prometaphase, and telophase)". Is anaphase an early mitotic cell for this Figure?
- 5) Fig R10: Cell cycle profile of NUMA1 MS2 and Sun tag shows small reduction of cells in G2/M (23.9% control vs 20.3% NUMA1 MS2 and 20.5% NUMA1 Sun tag). How the authors clarify that together with the opposite BAC effect for NUMA1 (see comment 1).

Reviewer #3 (Remarks to the Author):

Safieddine et al. answered most of my concerns and modified the manuscript accordingly. This is now a stronger manuscript that will be valuable in the field. I just have few minor points that need to be addressed in the text prior to publication.

Minor comments

Regarding the co-detection between translation factors and centrosomal mRNAs, I appreciate the efforts of the authors and understand how challenging are smFISH-IF experiments. I think that centrosomal localization of translation factors that was showed in Chouaib et al. is another important piece of evidence of local translation and I would suggest including this in the text either in the introduction or discussion.

Figure 3. Please, add scale bar to the cycloheximide treatment.

Figure legend 3 (p 39, line 861). Please, correct the typo: 'nul hypothesis' to 'null hypothesis'.

Reviewer #1 (Remarks to the Author):

I am satisfied with the significant efforts put forth by the authors to address reviewer concerns and now recommend acceptance of the manuscript.

We thank the reviewer for investing their time and expertise in reviewing our manuscript.

Reviewer #2 (Remarks to the Author):

The authors added a significant amount of new experiments, which strongly improved the manuscript, in particular new Figure 3. I have few comments regarding the new figures that would be good to discuss and address:

We are glad the reviewer appreciates the amount of added data, and clarify the raised points below:

1) Fig R4: there is a slight increase in G2/M population of cells in NUMA1 BAC cells, which is also obvious from the cell cycle profile. This suggests that introduction of BAC for NUMA1 might affect mitosis (24% Bac NUMA1 vs 19% controls).

This slight increase in G2M may be due to the expression of NUMA1-GFP, however it is also within measurement variability and could also be due to random clonal variations. In any case, it is difficult to imagine a scenario where this slight increase in G2M causes cells to actively traffic NUMA1 polysomes with a dedicated mechanism. In addition, looking at endogenous NUMA1 mRNAs in HeLa cells that do not express the BAC, we see similar localization of NUMA1 mRNA on centrosomes during mitosis (Fig. 8 for example).

2) Fig R6A: It seems like the metaphase cells showed in this image after emetine is bigger compared to the control metaphase. Are you sure the scale is the same? There is also a typo error, it should be written emetine, not ementine.

We thank the reviewer for pointing this out. We apologize for the change in zoom for this particular cell.

3) Fig R8: the authors treated the cells with centrinone and concluded that there is no effect of the centrinone on centrosomal mRNA localisation. Again, representing the cells in the

same stage of mitosis would be better despite the fact that ASPM mRNA levels are similar from prophase to anaphase (Fig 2 A, B). The same for NUMA1 mRNA, as its mRNA is high in prophase but goes down in prometaphase (Fig 2 C, D). From the representative images it looks like that centrinone reduced the signal of NUMA1 and ASPM mRNA on the centrosome.

While we agree that it would be best to represent cells of the same mitotic stage, as the reviewer mentioned for ASPM mRNA localization does not significantly vary across different stages of mitosis. Since despite treating cells with centrinone, we still see centrosomes with an IF, we think that it would be hazardous to draw solid conclusions from this kind of experiment. Moreover, the small effect observed in that cell is not consistently observed across cells or across different mRNAs.

4) Fig R9B: It would be better to show the same stage (metaphase) for ASPM mRNA after Cytochalasin D treatment. The authors should clarify what are the early mitotic cells counted for this graph as they stated on page 10 for Figure 2 : "Here, we found that ASPM uniquely localizes during all 5 mitotic phases, while NUMA1 localizes only during early mitosis (prophase and prometaphase), and HMMR during early and late mitosis (prophase, prometaphase, and telophase)". Is anaphase an early mitotic cell for this Figure?

We apologize for the ambiguity in the legend of Fig. R9. In this experiment, early mitotic cells (prophase and prometaphase) were counted for NUMA1 and HMMR, while all mitotic cells were considered for ASPM. This was done since in order to test whether Cytochalasin D de-localizes the mRNA, it is necessary to select a mitotic phase in which the mRNA naturally localizes.

5) Fig R10: Cell cycle profile of NUMA1 MS2 and Sun tag shows small reduction of cells in G2/M (23.9% control vs 20.3% NUMA1 MS2 and 20.5% NUMA1 SunTag. How the authors clarify that together with the opposite BAC effect for NUMA1 (see comment 1).

NUMA1 MS2 and NUMA1 SunTag clones are the result of CRISPR-Cas9 gene insertions. It is widely accepted that clonal selection can cause small changes in cell cycle profiles as observed here. Note that many experiments were carried out in the CRISPR clones and parental WT cells (drugs, localization profiles, etc.), and show similar results.

Reviewer #3 (Remarks to the Author):

Safieddine et al. answered most of my concerns and modified the manuscript accordingly. This is now a stronger manuscript that will be valuable in the field. I just have few minor points that need to be addressed in the text prior to publication.

Minor comments

Regarding the co-detection between translation factors and centrosomal mRNAs, I appreciate the efforts of the authors and understand how challenging are smFISH-IF experiments. I think that centrosomal localization of translation factors that was showed in Chouaib et al. is another important piece of evidence of local translation and I would suggest including this in the text either in the introduction or discussion.

Figure 3. Please, add scale bar to the cycloheximide treatment.

Figure legend 3 (p 39, line 861). Please, correct the typo: 'null hypothesis' to 'null hypothesis';

We thank the reviewer for their careful reading of the manuscript and pointing these out. We now corrected the typo, added scale bars, and referenced the centrosomal localization of translation factors.

We also appreciate the fact that the reviewer understands the complexity of smFISH-IF experiments.